

# The 2018 west-central European drought projected in a warmer climate: how much drier can it get?

Emma E. Aalbers[1,2], Erik van Meijgaard[1], Geert Lenderink[1], Hylke de Vries[1], Bart J. J. M. van den Hurk[2,3]

[1]Royal Netherlands Meteorological Institute (KNMI), PO Box 201, 3730 AE, De Bilt, Netherlands
[2]Institute for Environmental Studies (IVM), VU University Amsterdam, Netherlands
[3]Deltares, Delft, The Netherlands

*Correspondence to* emma.e.aalbers@gmail.com

**Abstract.** Projections of changes in extreme droughts under future climate conditions are associated with large uncertainties, owing to the complex genesis of droughts and large model uncertainty in the atmospheric dynamics. In this study we investigate the impact of global warming on soil moisture drought severity in west-central Europe by employing pseudo-global warming (PGW) experiments, which project the 1980-2020 period in a globally warmer world. The future analogues of present-day drought episodes allow investigation of changes in drought severity conditional on the historic day-to-day evolution of the atmospheric circulation.

The 2018 west-central European drought is the most severe drought in the 1980-2020 reference period in this region. Under 1.5°C, 2°C and 3°C global warming, this drought episode experiences strongly enhanced summer temperatures, but a fairly modest soil moisture drying response compared to the change in climatology. This is primarily because evaporation is already strongly moisture-constrained during present-day conditions, limiting the increase in evaporation and thus the modulation of the temperature response under PGW. Increasing precipitation in winter, spring and autumn limit or prevent an earlier drought onset and duration. Nevertheless, the drought severity, defined as the cumulative soil moisture deficit volume, increases considerably, with 20% to 39% under 2°C warming.

The extreme drought frequency in the 1980-2020 period strongly increases under 2°C warming. Several years without noticeable droughts under present-day conditions show very strong drying and warming. This results in an increase in 2003-like drought occurrences, compounding with local summer temperature increases considerably above 2°C.

Even without taking into account a (potentially large) dynamical response to climate change, drought risk in west-central Europe is strongly enhanced under global warming. Owing to increases in drought frequency, severity and compounding heat, a reduction in recovery times between drought episodes is expected to occur. Our physical climate storyline provides evidence complementing conventional large-ensemble approaches, and is intended to contribute to the formulation of effective adaptation strategies.

## 1    Introduction

The impact of recent west-central European droughts and heat waves on society and nature (Vogel et al., 2019, Rösner et al., 2019, Kramer et al., 2019, Schuldt et al., 2020, Beillouin et al., 2020, Bastos et al. 2021, Krikken et al., 2021) once again triggered questions regarding the role of climate change in the occurrence and extremity of drought events (Kornhuber et al., 2019, Yiou et al., 2020, Philip et al., 2020, Zscheischler and Fischer, 2020) and on what to expect under continuing global warming (Toreti et al., 2019, Kornhuber et al., 2019, Hari et al., 2020).



The 2018 growing season was the compound hottest-and-driest ever recorded in west-central Europe (Toreti et al., 2019, Zscheischler and Fischer, 2020), owing to a sequence of anomalously persistent high pressure systems over eastern, northern, and central Europe between April and October (Bissolli, 2019, Sluijter et al., 2018), associated with large-scale atmospheric subsidence, clear sky conditions and generally low relative humidity and moisture advection (Sousa et al., 2017, 2018),
against the background of globally increasing temperatures (Philip et al., 2020, Vogel et al., 2019). Temperatures were anomalously high over almost the entire European continent (Vogel et al., 2019, Kornhuber et al., 2019), but the precipitation deficit was particularly intense and long lasting in west-central Europe. In this region the deficit built up from April/May until November, only intermittently interrupted by intense but small-scale short-duration rainfall events (Bissolli, 2019, Sluijter et al., 2018). This led to soil desiccation and extremely low groundwater tables (Brakkee et al., 2022) and river
discharge (Brunner et al., 2019, Kramer et al., 2019) in the west-central European river basins. The consecutive years 2019 and 2020 were characterized by record-braking heatwaves (Vautard et al., 2020, Sousa et al., 2020) and anomalously dry conditions as well (Hari et al., 2020, Bastos et al., 2021, Bissolli, 2020, 2021, Rakovec et al., 2022, Van der Wiel et al., 2022). And again, in 2022, heat waves and severe and particularly widespread drought conditions affected Europe (Toreti et al., 2022). At the time of writing, soil moisture deficits, river water levels and river discharge approached or exceeded 2018
levels in several west-central European river basins, with reported impacts on ecology, agriculture and shipping (Toreti et al., 2022, WMCN-LCW, 2022, BfG, 2022).

Although the probability of heat waves in this region is demonstrated to have increased in response to anthropogenic climate change (Stott et al., 2004, Vogel et al., 2019, Vautard et al., 2020), the attribution of extreme drought conditions is complicated by the complexity of processes contributing to wide-spread drought conditions (Trenberth et al., 2014), while
deriving robust statistics is hampered by the scarcity of independent drought events owing to their long timescale and large spatial scale. Intense drought conditions are governed by persistent patterns of atmospheric circulation with low moisture advection into the region of interest. Trends over recent years suggest increases in the frequency and/or persistence of such circulation conditions (Coumou et al., 2014, Kornhuber et al., 2019), but observed circulation-related changes are generally dominated by natural variability (Shepherd, 2014) and there are no significant long-term trends in meteorological
(precipitation) drought events in this region (Gudmundsson and Seneviratne, 2016, Hanel et al., 2018, Manning et al., 2019, Spinoni et al, 2019, Philip et al., 2020, Gutiérrez et al., 2021). Nevertheless, observation- and model-based studies find decreasing trends in summer water availability (SPEI, precipitation minus evaporation) (Spinoni et al. 2019, Padrón et al., 2020), increases in the frequency and/or severity of soil moisture droughts (Hanel et al., 2018, Philip et al., 2020) as well as in long-duration compound hot-and-dry events (Manning et al., 2019). This is owing to increasing trends in atmospheric
evaporative demand with global warming in the predominantly energy-constrained evaporation regime in west-central Europe.

Under further increasing greenhouse-gas concentrations, climate projections agree on a general pattern of year-round decreasing precipitation in the Mediterranean and increasing precipitation in northern Europe, with the drying/wetting transition zone shifting north in summer under higher levels of global warming. For west-central Europe precipitation
increases are projected for winter and autumn, while smaller increases or small decreases are projected for spring and summer (Jacob et al., 2014, Aalbers et al., 2018, Coppola et al., 2021, Gutiérrez et al. 2021). Soil moisture is projected to further decrease, with strongest responses in summer and autumn (Ruosteenoja et al., 2018, Van der Linden et al., 2019) and studies based on large model ensembles show increases in the frequency and severity of (multi-year) drought episodes (Samaniego et al., 2018, Toreti et al., 2019, Hari et al., 2020). The magnitude and direction of the precipitation changes and the magnitude
and timing of the soil moisture drying response are uncertain, and depend on e.g. the climate model resolution and generation (Jacob et al., 2014, Coppola et al., 2021, Van der Linden et al., 2019), biases in the mean climate state in the reference period, and the ability of climate models to realistically represent land-surface-atmosphere coupling (Orth et al., 2016, Van der



Linden et al., 2019, Vogel et al., 2018, Selten et al., 2020) and atmospheric dynamics (Shepherd, 2014, Woollings et al., 2018).

In this study the contribution of global warming to the increase in drought severity and frequency is being addressed by putting the 1980-2020 historical period in the context of a globally warmer world. We hereby follow a pseudo global warming (PGW) approach (Schär et al., 1996), by which we create "future weather analogues" (Hazeleger et al., 2015, Shepherd, 2018, Shepherd et al. 2019, Sillmann et al., 2021, Van der Wiel et al., 2021, Wehrli et al., 2020) of present-day summers. The present-day simulations are performed with a regional climate model (RCM) forced with reanalysis data. For

the PGW simulations we essentially re-run the simulations, but perturb the atmospheric and oceanic forcing data with climate change information from global climate model (GCM) projections. This approach has been shown to capture a large part of the climate change signal, optimizing the signal-to-noise ratio by suppressing responses from large-scale atmospheric circulation variability (Brogli et al., 2019, De Vries et al., 2022, Lenderink et al., 2022). As such, changes in droughts can be directly related to real world events and their societal impact, which make the results very tangible and therewith useful

for climate change communication. Another advantage is that the reference climate state, which can have a large impact on drought evolution, is not affected by biases in a GCM since it is based on reanalysis data, thus avoiding one source of uncertainty in future projections.

We focus our analysis on the 2018 drought episode for its recent occurrence and severe impact. Based on the present-day simulations we first explore the atmospheric drivers and soil moisture evolution of the 2018 event under present-day

conditions. We repeat this analysis with the PGW simulations, with perturbations derived from three different GCM projections and for several levels of global warming, to diagnose the response in atmospheric drivers, the soil moisture evolution and the severity of the 2018 drought event. Additionally we evaluate the position of this 2018 event in the 1980-2020 period, both for present-day and for future conditions under a single warming level.

The purpose of this work is to provide robust, physically consistent scenarios of what global warming entails for extreme

droughts, and for the full range of wet to moderately dry years that occurred in the historical record. It is intended to complement projections of changes in drought risk derived with the conventional large-ensembles approaches, giving an explicit reference to collectively experienced real world events.

## 2 Model and methods

### 2.1 Regional climate model

All simulations are performed with the RCM KNMI-RACMO2 (Van Meijgaard et al., 2012), run at 12km resolution, with 40 vertical model levels. External forcings for aerosols and greenhouse gases have been implemented according to CMIP5 prescriptions (Collins et al., 2013). RACMO2 uses the land surface scheme HTESSEL (Balsamo et al., 2009), which employs four soil layers with a total depth of 2.9 m. At the bottom of the soil column, boundary conditions are specified as zero-heat flux and free drainage. Each land-grid cell includes separate tiles for high and low vegetation (16 vegetation types), bare

soil, snow on low vegetation/bare soil, snow beneath high vegetation and intercepted water, for which the energy and water balances are solved individually. The tile fractions vary with interception storage and snow cover. The vegetation cover (leaf area index) follows a fixed annual cycle. The model domain is centered over west-central Europe, and covers most of Europe.

### 2.2 Experimental setup

The analyses are based on two sets of RCM simulations: present-day simulations (REF) and pseudo-global-warming (PGW)

simulations. Both sets include a single climate run covering the period 1980-2017 for present-day conditions (climREF) and





**Table 1. Model simulations. The data source for the initial land surface conditions (Land surface init.), and the sea surface and initial and lateral atmospheric boundary conditions (Sea & atm. init. & bound.) are indicated. $\Delta_{nK}$ = perturbations for $n$ (°C/K) global warming derived from *GCM* projections, EC = EC-EARTH , HAD = HadGEM2-ES, MPI = MPI-ESM-LR.**

| | Period | Start date | **Present-day simulations (REF)** | | | | **PGW simulations (+$n$K\|$GCM$)** $n$ = 1.5, 2, 3; $GCM$ = EC, HAD, MPI | | | |
| | | | Name | Member (mb) | Land surface init. | Sea & atm. init. & bound. | Name | Member (mb) | Land surface init. | Sea & atm. init. & bound. |
|---|---|---|---|---|---|---|---|---|---|---|
| **clim** | 1979-2017 | 1 Jan 1979 | **climREF** | - | ERA5 | ERA5 | **clim+2K\|$GCM$** | - | ERA5 + $\Delta_{2K}$ | ERA5 + $\Delta_{2K}$ |
| **2018** | 2018 -2020 | 1 Jan 2018 <br> 6 Jan 2018 <br> ⋮ <br> 20 Feb 2018 | **2018REF** | 1 <br> 2 <br> ⋮ <br> 11 | climREF <br><br> 2018REF, mb1 | ERA5 | **2018+$n$K\|$GCM$** | 1 <br> 2 <br> ⋮ <br> 11 | 2018REF, mb1 + $\Delta_{nK}$ <br><br> 2018+$n$K\|$GCM$ , mb1 | ERA5 + $\Delta_{nK}$ |

2°C global warming (clim+2K) and an 11-member initial-condition ensemble for the period 2018-2020 for present-day conditions (2018REF) and 1.5°C, 2°C and 3°C global warming (2018+$n$K; $n$ = 1.5, 2, 3). By creating an ensemble, random small-scale variations in the weather (i.e. due to internal variability within the RCM domain) are sampled, increasing the robustness of the assessment of future changes. This is especially relevant for the analysis of short-term weather/climate events. Tab. 1 provides an overview of all simulations. The simulations are detailed in the following subsections.

*Present-day simulations* First, RACMO is run continuously over the period January 1st 1979 – January 1st 2018, with initial conditions and lateral and sea surface boundary conditions from the ERA5 reanalysis dataset (Hersbach et al., 2020). The sea surface and lateral boundary conditions are updated every 3 hours. The first year is used as spin-up, leaving the period 1980-2017 as the reference period for the present-day climate (climREF). The 11-member ensemble for 2018-2020 (2018REF) is created by running RACMO eleven times over the period 2018-2020, reinitializing the atmospheric state to 130 the ERA5 reanalysis at January 1st for member 1, January 6th for member 2, up to February 20th for member 11. Unless indicated otherwise, throughout this paper analyses for 2018-2020 are based on the ensemble mean values.

*PGW-simulations* To examine the impact of global warming, all simulations are rerun, but with perturbed initial land state (soil moisture, soil temperature, snow cover), sea surface (temperature and sea ice extent) and lateral boundary conditions (temperature, humidity, geopotential height and wind), representing the change in the mean climate state in a globally 135 warmer world. We impose a single 2°C global warming to the 1979-2020 simulation (clim+2K). To examine the sensitivity of the 2018 response to the warming level we impose three different global warming levels to the 11-member 2018 ensemble (2018+1.5K, 2018+2K and 2018+3K). The perturbations are determined from GCM projections as the 3-dimensional monthly mean climate change signal in the 30-year period in which the target global warming level with respect to present-day conditions (1991-2020) is reached (Tab. A1). Therewith we capture a large part of the climate change signal, including 140 mean changes in the vertical temperature, humidity and wind profiles and in the mean circulation. However, the day-to-day evolution of the synoptic-scale circulation, i.e. the sequence of weather systems entering the model domain, in the PGW simulations remains essentially determined by the reanalysis forcing and is therefore very similar to the sequence seen in the present-day simulation.

Since climate models differ in their regional climate response, we apply perturbations derived from three different GCM 145 initial-condition ensembles: a 16-member EC-EARTH v2.3 (Hazeleger et al., 2010) ensemble produced at KNMI, a 4-member HadGEM2-ES (Collins et al., 2011) ensemble and a 3-member MPI-ESM-LR (Giorgetta et al., 2013) ensemble from the CMIP5 archive (Taylor et al., 2012), referred to as respectively EC, HAD and MPI. The perturbations are derived from the ensemble means of the initial-condition GCM ensembles rather than from a single simulation per GCM to obtain a more robust estimate of the forced climate response (Deser et al., 2010, Fischer et al., 2014, Aalbers et al., 2018). All GCM 150 ensembles are run under the RCP8.5 emission scenario. The main characteristics of the three perturbation sets are shown in





appendix A. Obviously there are large similarities in the climate change response between the three ensembles, but details like e.g. the response in the spatial pressure gradient and the shape of the vertical temperature response are different. In terms of regional warming and drying, differences are most pronounced in spring and summer. HAD exhibits the strongest warming in spring, MPI shows the strongest warming and drying in summer.

**2.3    Model evaluation**

The simulated 2-m temperature and precipitation are evaluated against the gridded observational dataset E-OBS v20.0e (Cornes et al., 2018).

**2.4    Indicators and variables**

We identify soil moisture drought conditions based on the exceedance of a seasonally varying threshold of the soil wetness

index (SWI, (-)) of the top 1 m of the soil. The top 1 m of the soil is where – in HTESSEL – vegetation has the highest root density and where water deficiencies have the strongest link to agricultural drought (Seneviratne et al., 2012). The SWI is the fraction of plant available water in the soil, defined as the soil moisture availability ($\theta$, (mm)) scaled between field capacity ($\theta_{fc}$) and permanent wilting point ($\theta_{pwp}$) (Eq. 1). The SWI is better suited for aggregation over areas with different soil types than $\theta$ itself. $\theta_{fc}$ and $\theta_{pwp}$ are fixed characteristics per grid cell.

$$SWI = \frac{\theta - \theta_{pwp}}{\theta_{fc} - \theta_{pwp}}$$    (1)

A soil moisture drought event is defined as the consecutive period in which the soil moisture conditions are drier than the 5th percentile threshold of the 1980-2017 SWI climatology (SWI<SWI$_{5th}$) (Trenberth et al. 2014). SWI$_{5th}$ is calculated for every calendar day based on 14-day smoothed SWI values. We apply the same drought threshold for the present-day and PGW conditions, to benchmark the warming induced changes to present-day conditions. We express the drought severity in

terms of the drought deficit volume (unit mm d), which integrates drought duration (d) and drought intensity (mm), comparable to e.g. Yevjevich (1967) and Brunner et al. (2019). It is calculated as the accumulated difference between $\theta_{5th}$ and $\theta$ over the drought episode. The drought intensity is defined as the drought deficit volume divided by the drought duration.

In the analyses we use the atmospheric evaporative demand synonymously with potential evaporation ($E_p$), the evaporation

that would take place assuming unconstrained conditions with respect to soil moisture availability and vapor pressure deficit. The computation of evaporation in HTESSEL uses a resistance approach based on Jarvis (1976), for each individual land cover tile, see ECMWF (2009). To obtain a potential evaporation measure that is fully consistent with the simulated actual evaporation, it is diagnosed in a parallel calculation within RACMO2, using the prevailing atmospheric conditions, but with resistance functions accounting for soil moisture availability and vapor pressure deficit set at a value representing

unconstrained conditions. See for details appendix B.

**2.5    Study area**

We focus on the larger river basins in west-central Europe that discharge in the North Sea, namely the Rhine, Meuse, Scheldt, Ems, Weser and Elbe. These river basins are part of the area where the 2018 soil moisture drought episode was most severe and lasted longest, as shown in Fig. 1.



## 3 The 2018 drought episode in the present-day climate

We first present the main characteristics of the simulated 2018 drought episode, and briefly discuss the evaluation of temperature and precipitation against observations. In Fig. 1a maps of the simulated 2018 seasonal anomaly in 500 hPa geopotential height (contours), temperature, precipitation, evaporation and soil moisture are shown for April – June (AMJ), July – September (JAS) and October – December (OND). Anomalies are calculated from 2018 in the 2018REF simulation relative to the 1980-2017 period (climREF). Time series of these variables averaged over the west-central European river basins are shown in Fig. 1b, with observed temperature and precipitation delineated in orange.

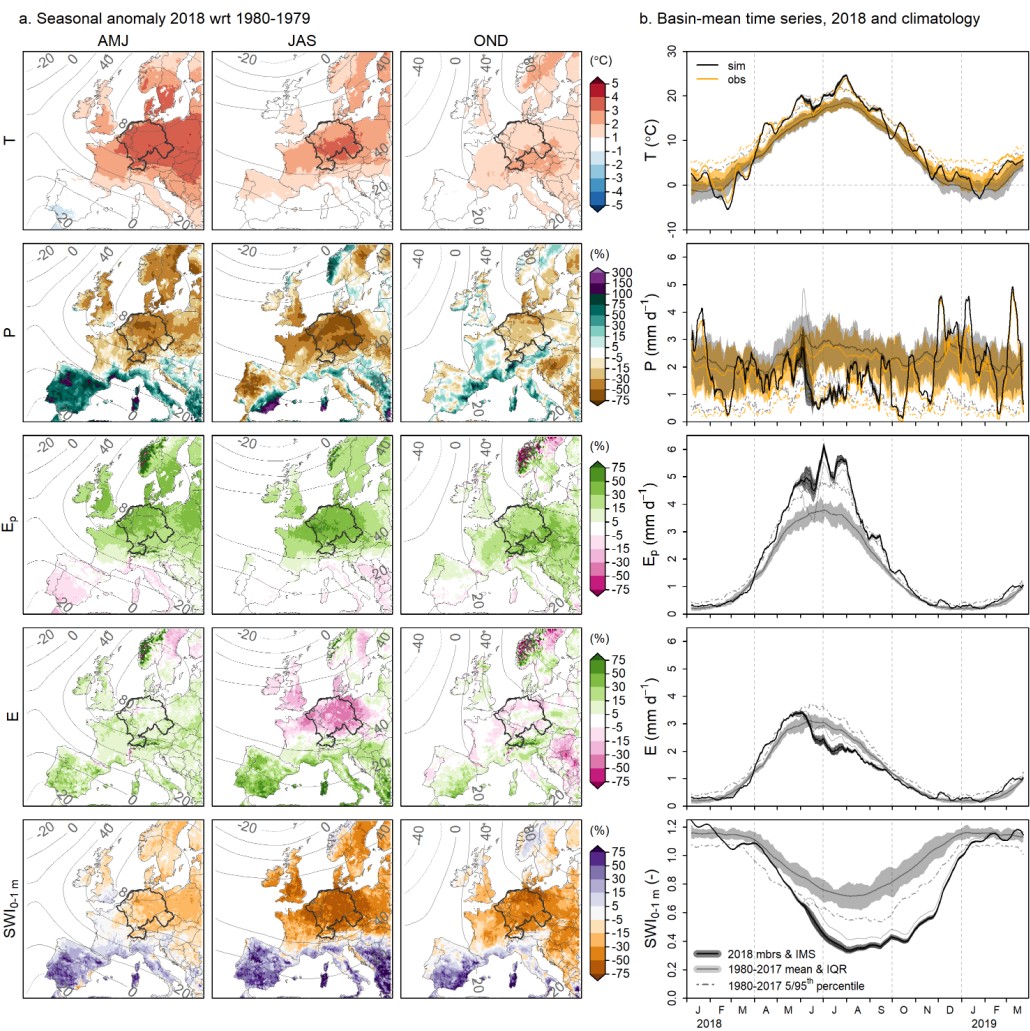

Figure 1: The 2018 drought episode. a) Maps of the simulated seasonal mean anomaly in 500 hPa geopotential height (contours) and (top to bottom): near surface-temperature ($T$), precipitation ($P$), evaporation ($E$) and top 1 m soil wetness index ($SWI_{0-1m}$), for April – June (AMJ), July – September (JAS) and October – December (OND). Data are masked over sea for visibility. The west-central European river basins are marked in black. b) Time series of the simulated basin-mean (top to bottom) temperature, precipitation, (potential) evaporation ($E_{(p)}$) and top 1 m soil wetness index for January 2018 – March 2019 (black) and climatology (grey). Observed (E-OBS v20.0) temperature and precipitation are shown along in orange for 2018 (line) and climatology (shading). Time series are smoothed with a 14-day running mean. The dark lines and shading show the 2018REF ensemble members and the inter-member spread (IMS, mean +/- 1 standard deviation). The lighter line and shading and the dashed line depict respectively the 1980-2017 mean, interquartile range (IQR), and the 5th ($P$ and $SWI$) or 95th ($T$, $E$, $E_p$) percentile.



The high pressure anomalies in (late) spring (AMJ) and summer (JAS) clearly co-occur with the large positive temperature anomalies and high precipitation deficits. Averaged over the west-central European river basins the simulated (observed) temperature anomaly is +3.1ºC (+2.5ºC) over the growing season (April to September), and temperatures exceed the 95th percentile during several episodes. Most noteworthy are 8 – 22 April, with a 15-day mean temperature anomaly of +6.5ºC (+6.0ºC), and 22 July – 8 August, with an 18-day mean anomaly of +5.9ºC (+5.1ºC). The latter period was indeed classified as heat wave in the individual countries (Yiou et al. 2020, Sluijter et al. 2018, Vogel et al. 2019, Bissolli et al. 2019). Apart from a cold bias in winter, the basin-mean simulated absolute temperatures are fairly accurate, with a small underestimation with respect to the observed 1980-2017 mean temperature in the growing season (-0.3ºC) and overestimation of the extreme conditions of 2018 temperatures (+0.3ºC) in most members of the 2018REF ensemble.

Basin-mean precipitation is anomalously low in each month from February to November. Averaged over the growing season, the simulated (observed) basin-mean precipitation anomaly is -1.1 mm/d or -41% (-37%), with largest, basin-wide deficits in June (-56% (-43%)) and in July (-64% (-64%)). Mean precipitation is overestimated compared to the observations with on average 0.2 mm/day, both for the climatology and 2018.

Under prevailing conditions of clear skies, high solar radiation, high temperatures and increasingly dry air, the response in the atmospheric evaporative demand is substantial (+1 mm/d or +35% over the growing season). Also the actual evaporation is anomalously high from April up until beginning June, modulating the near-surface temperatures. However, it cannot keep up with the rise in atmospheric evaporative demand, owing to quickly increasing soil and canopy resistance against evaporation in response to decreasing relative humidity and soil moisture availability, and has below normal values from mid-June to October. As a consequence, the sensible heat flux strongly increases (not shown), which corresponds to an amplified rise in summer and autumn near-surface temperatures. Averaged over the growing season, the actual evaporation is slightly smaller than normal (-0.1 mm/day or -6%).

The resulting extremity of the 2018 soil moisture drought is clearly reflected (Fig. 1, bottom row). Anomalously low soil moisture levels occur in large parts of central and northern Europe, but, consistent with the persistent precipitation deficits, conditions are most severe and persistent in west-central Europe. Averaged over the west-central European river basins the soil moisture conditions are around normal at the start of the growing season, owing to low temperatures and evaporation in March. Soils steadily deplete from April onwards, reach severely dry conditions (exceeding the 5th percentile) in the second half of May and are lowest in early August. Soil moisture levels remain very low throughout the growing season up to end October. This is when precipitation starts to exceed the evaporation and soil moisture replenishes, reaching the 5th percentile threshold in the beginning of January 2019, after nearly 8 months of severely dry conditions. By then, the soil moisture deficit volume has accumulated to 8240 mm d, with a mean drought intensity of 36 mm. In Sect. 5.1 and Fig. 6 we will show the extremity of this number compared to other drought episodes in the 1980-2020 period. Normal soil moisture levels in the top 1 m of the soil are reached early February 2019. For deeper soil layers the winter precipitation is insufficient to fully replenish the soils to normal levels, and the anomalously dry conditions persist throughout 2019 (not shown).

The overestimation of precipitation could imply an overestimation of the soil moisture levels. On the other hand, the overestimation of the 2018 summer temperature likely leads to a dry bias in soil moisture. The amount of inter-member spread in the 2018REF ensemble (natural variability generated within the RCM model domain) is found to be considerable in some periods, as seen in Fig. 1b (dark shading). The spread is largest in the period end May-early June when variability in the location and intensity of precipitation bearing systems induces relatively strong variability in wetness and temperature. While the ensemble spread in temperature is relatively short-lived, the ensemble spread in soil moisture reduces more slowly over summer. One ensemble member receives much higher precipitation amounts in May – early June, as well as in July



and August. Evaporation in this member is consequently relatively high throughout summer, and the temperature is 1.1°C (June) to 0.3°C (September) lower than the ensemble mean, closer to the observations. Apart from model biases and natural variability, differences between actual and simulated atmospheric and soil conditions are possibly related to interactions between soil moisture and groundwater in especially the low-lying coastal areas, which are not taken into account in HTESSEL.

## 4 Response to Pseudo Global Warming

### 4.1 Climatological mean response to 2°C warming

We next present the climatological mean response to a 2°C warming, to provide context to the 2018 response. Figure 2 shows the seasonal response patterns in geopotential height, near-surface temperature, precipitation, (potential) evaporation, and soil moisture over Europe for the EC-perturbed simulations. The annual cycle in the basin-mean response in these and additional variables is shown in Fig. 3. Results for the HAD- and MPI-perturbed simulations can be found in appendix C.

The spatial response patterns exhibit the well-known seasonally-varying warming and drying gradients over Europe (e.g. in the EURO-CORDEX ensemble (Coppola et al., 2021) and the RACMO-EC-EARTH initial-condition ensemble (Aalbers et al., 2018), showing that the PGW-simulations indeed capture the main characteristics of the full climate response (Brogli et al., 2019, De Vries et al. 2022). In spring, autumn and winter the warming gradient is oriented roughly northwest-southeast, with weakest warming above the British Isles and coastal regions adjacent to the Atlantic Ocean and North Sea (blue colors represent below 2°C warming). In summer, warming ranges between around +2.0°C over Scandinavia to around +3.0°C, locally +3.5°C in southern Europe. Averaged over the river basins the near-surface temperature response varies between +1.4°C in May and +2.6°C in August (Fig. 3a). Note that the inter-annual spread around the 1980-2017 mean response is rather large, especially in JAS, which will be discussed in Sect. 5.2.

The transition zone of increasing precipitation in the north and decreasing precipitation in the south is positioned just southwest of the west-central European river basins in spring and autumn and over the northeast of the basins in summer, yielding increasing precipitation in winter, autumn and early spring, and small decreases in summer for the basin-mean (Fig. 2b, 3e). This co-occurs with nearly constant relative humidity and increasing cloud cover in late autumn, winter and early spring, decreases in relative humidity and cloud cover in JJASO, and consequent increases in net surface solar radiation in this period (Fig. 3b,d). Under conditions of higher temperatures, and enhanced by the increase in solar radiation and decrease in relative humidity from late spring to late autumn, the atmospheric evaporative demand increases over land throughout the year (Fig. 2c, 3f). The present-day soil moisture regime in west-central Europe allows for year-through increases in actual evaporation in almost all years, with around potential rate in winter and early spring, but smaller than potential in JJASO (Fig. 2d, 3g) resulting in increases in the sensible heat flux in the latter period (Fig. 3c).

For the combined river-basin area, the increased evaporation and reduced summer precipitation lead to enhanced soil moisture depletion in late spring and summer, while in autumn and winter increases in precipitation result in a faster soil moisture replenishment. This feature of the response is amplified by reduced snowmelt in spring and a larger fraction of precipitation falling as rain in autumn (not shown). The resulting soil moisture levels in the top 1 m of the soil are around present-day or even wetter conditions in winter and early spring, but drier from mid-June to December, with a maximum drying response in September (Fig. 2e, 3h). In summer and autumn, the soil moisture availability in deeper layers and runoff decrease as well (not shown). However, the response in annual precipitation equals the response in annual evaporation, meaning that each winter soil moisture levels in all layers are restored to present-day levels, and decreases in summer runoff are compensated by increases in winter.




With the amplitude of the response and the position of the drying/wetting transition zone being dependent on the GCM, the MPI- and HAD-perturbed simulations give slightly different results, see Fig. C1 and C2. The drying/wetting transition zone is located further northeast in all seasons for both clim+2K|MPI and clim+2K|HAD. clim+2K|MPI shows a weaker

temperature response in spring, but much stronger drying and warming in JAS, consistent with a strong response in the geopotential height anomaly. The soil moisture depletion over the growing season in the west-central European river basins is consequently stronger, but so is the soil moisture replenishment in autumn and winter. clim+2K|HAD shows a stronger temperature and evaporation response in spring than the EC- and MPI-perturbed simulations, increases in precipitation are overall smaller and soil moisture levels are found to decrease earlier in spring.

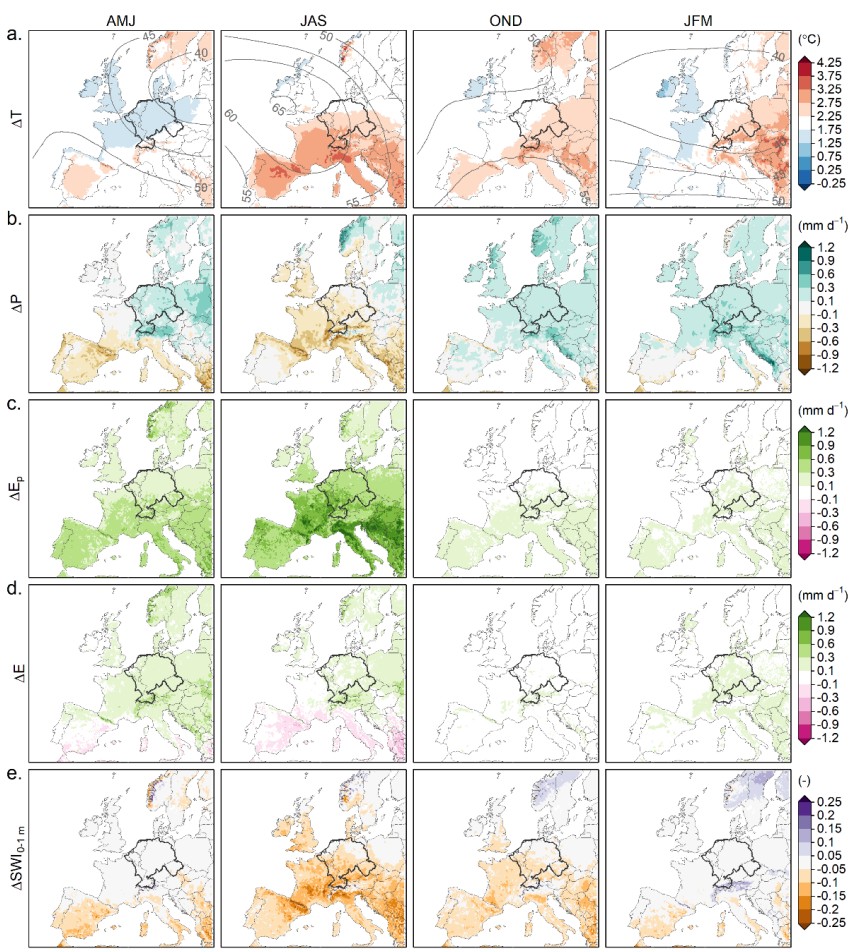

**Figure 2: Climatological mean response (1980-2017) to 2°C warming in a) the geopotential height at 500hPa (contours) and near-surface temperature (T, shading), b) precipitation (P), c) potential evaporation (Ep), d) evaporation (E) and e) the top 1m soil wetness index (SWI0-1m), averaged over April - June (AMJ), July - September (JAS), October - December (OND) and January - March (JFM). Results are based on climREF and clim+2K|EC.**




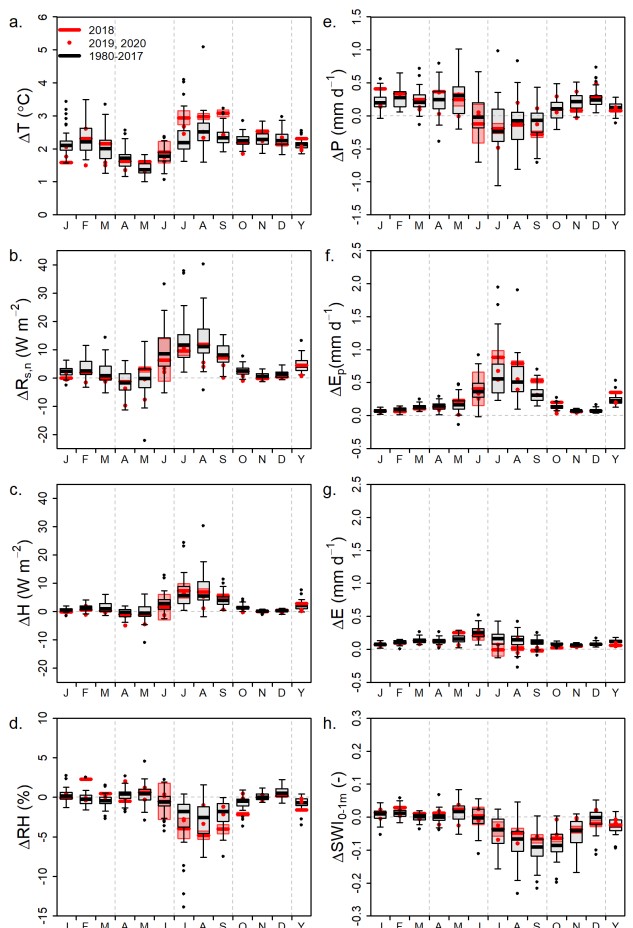

**Figure 3: Annual cycle in the basin-mean response to 2°C global warming for 1980-2020, for the EC-perturbed simulations in a) near-surface temperature ($T$), b) net solar radiation at the surface ($R_{s,n}$), c) sensible heat flux ($H$), d) near-surface relative humidity ($RH$), e) precipitation ($P$), f) potential evaporation ($E_p$), g) actual evaporation ($E$) and h) top 1 m SWI. The monthly (J-D) and annual (Y) response to 2°C warming are calculated for each year. Boxplots show the inter-annual distribution of the**
300 **response for the years 1980-2017, depicting the median (black bar), interquartile range (box), the total range with a maximum distance of 1.5 times the interquartile range outside the box (whiskers), and outliers (black dots). The red bars and box delineate the ensemble mean and inter-member spread (mean +/- 1 standard deviation ($\sigma_\Delta$)) of the response for 2018, the red dots the ensemble mean response for 2019 and 2020. Since the members in the 2018REF and 2018+2K ensembles are in principle independent, $\sigma_\Delta$ is calculated from the standard deviation ($\sigma$) in the present-day and +2K simulations, as $\sigma_\Delta =$**

$\sqrt{(\sigma^2_{present\,day} + \sigma^2_{2K})/2}.$





### 4.2    2018 response to 2°C warming

The response of the hot and exceptionally dry growing season of 2018 that unfolded under persistent conditions of atmospheric blocking is shown for the basin mean in Fig. 3 (red boxes). Maps of the 2018 response anomaly with respect to the climatological mean response are shown in Fig. 4 for 2018+2K|EC. Results for 2018+2K|HAD and MPI can be found in appendix C.

The 2018 response in winter and early spring preceding the blocking conditions is very similar to the climatological mean response in most variables and results in slightly wetter soil moisture conditions at the start of the growing season in April in 2018+2K|EC than in 2018REF. Also in spring, the 2018 soil moisture and circulation anomalies don't have a strong effect on the response. Consistent with the climatological mean response, precipitation is found to increase in April and May, with relatively strong increases in April. Apparently, the precipitation events originate from sources with sufficient moisture supply to sustain these increases. Note that the precipitation response is rather patchy (Fig. 4b), despite the application of the 11-member ensembles. Evaporation rises with more than potential rate in these months, but only partially compensates the precipitation increase. Despite a small decrease in snowmelt and increase in runoff (not shown), the top 1 m of the soil is slightly wetter until mid-June in 2018+2K|EC than in 2018REF.

From mid-June onwards strong deviations from the climatological mean response occur for the temperature, relative humidity, atmospheric evaporative demand and evaporation responses, exceeding the 25th-75th percentile range of 1980-2017 (Fig. 3a,d,f,g and 4a,c,d). Decreases in precipitation (June - September) and the weak evaporation response (July - September) show that sources of moisture are even more limited in a 2°C warming scenario. Precipitation in this period originates from predominantly continental sources (Benedict et al., 2021), and the evaporation response is moisture-constrained throughout Europe (Fig. 4c,d).

From July to September the temperature response over the basin area and surroundings is amplified compared to the climatological mean response (+3.0°C over JAS 2018 compared to +2.4°C for the climatology in the basin-area). This response anomaly correlates with the anomalously low actual evaporation response pattern. Within the basin, evaporation barely increases or even decreases in the period July - October, and co-occurs with a further decrease in relative humidity, a modest increase in the sensible heat flux (Fig. 3c) and increase in near-surface temperature. Note that the increase in solar radiation is relatively small in June and July (Fig. 3b), given the predominantly clear-sky conditions in 2018REF, and that the increase in the sensible heat flux is only slightly larger than the climatological response. Increases in heat advection due to stronger warming in upwind regions or enhanced warming through subsidence may play a role in the amplified warming as well.

Since the response in summer evaporation in the west-central European river basins is close to zero, the JAS soil moisture response is small compared to most other years in the 1980-2020 period (Fig. 3h), and is almost completely determined by the decrease in precipitation. The pattern of the soil moisture response anomaly strongly correlates with the precipitation response anomaly in this period (Fig. 4e). Percolation to deeper soil layers and runoff decrease in this period as well (not shown). In autumn and winter, moderate precipitation increases replenish the soils to 2018REF levels in December/January, in the top 1 m of the soil and in deeper layers.

### 4.3    Sensitivity of the 2018 response to the level of global warming and GCM perturbations

In Fig. 5 we show the 2018 basin-mean timeseries of the anomaly in near-surface temperature and top 1m SWI (Fig. 5a), and the hydrological budget changes over AMJ and JAS (Fig. 5b) for all warming levels and PGW simulations (EC, MPI and HAD).




In the EC-perturbed simulations, the temperature response is fairly linear with global warming under the increasingly moisture-constrained conditions, with 2.2°C, 3.0°C and 4.3°C warming during the July/August heatwave under respectively 1.5°C, 2°C and 3°C global warming, and 1.7°C, 2.3°C and 3.4°C warming for the growing season mean. This yields temperature anomalies during the July/August heatwave (i.e. deviations with respect to climREF) of +8.1°C, +8.9°C and

+10.2°C, compared to +5.9°C under present-day conditions. The soil moisture depletion over the growing season increases with higher levels of global warming, but only from mid-June onwards. In AMJ, precipitation increases are stronger under 3°C warming than under 1.5°C, but so are the evaporation increase, snowmelt decrease and runoff increase, resulting in the almost zero change in soil moisture depletion in the top 1 m of the soil over AMJ for higher levels of global warming. The increase in soil moisture depletion over JAS for higher levels of global warming is mainly driven by stronger decreases in

JAS precipitation. Although the JAS evaporative demand increases with higher levels of global warming, actual evaporation does not increase or only very weakly. Note that the soil moisture depletion in deeper soil layers is more pronounced and occurs throughout the growing season (Fig. 5b, red bar).

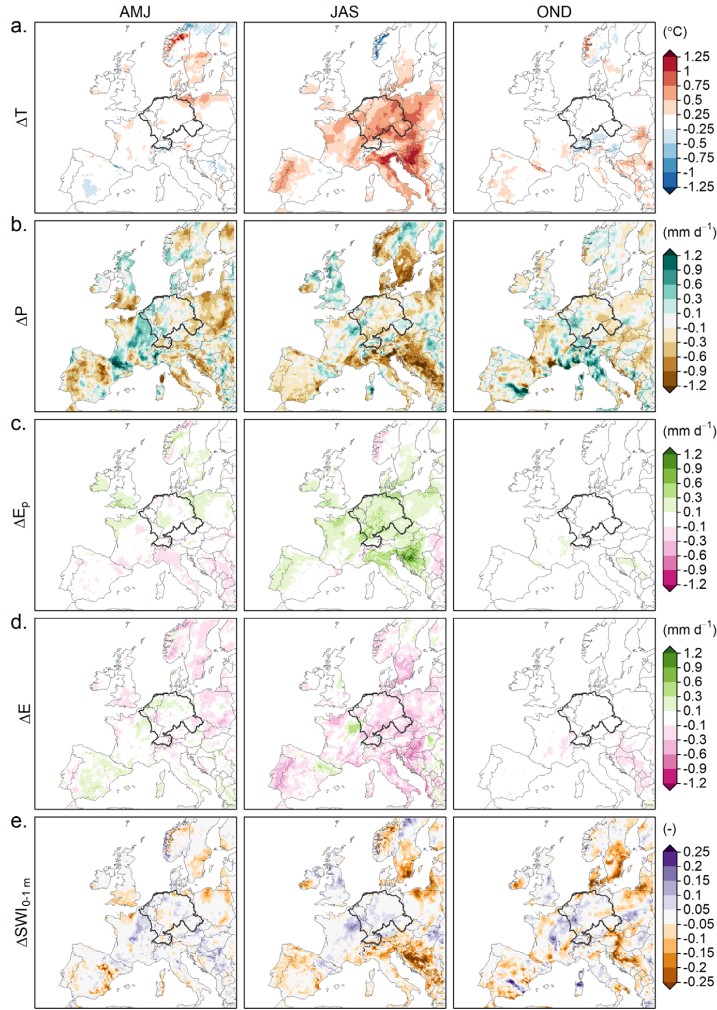

**Figure 4: Difference between the 2018 and climatological mean (1980-2017) response to 2°C warming (2018+2K|EC - 2018REF)**
**- (clim+2K|EC-climREF) in a) temperature, b) precipitation, c,d) (potential) evaporation and e) SWI in the top 1 m of the soil.**





In the MPI-perturbed simulations for 2018, the soil moisture response is fairly similar to the EC-perturbed simulations in spring, despite a weaker increase in AMJ precipitation, which is compensated by a weaker increase in evaporation. Also in summer and autumn the soil moisture drying is fairly similar under 1.5°C and 2°C warming, as is the response in the hydrological budget terms. Under 3°C warming the soil moisture drying is more pronounced owing to a stronger decrease 365 in precipitation. JAS evaporation decreases, and the JAS temperature response is amplified compared to 1.5°C and 2°C warming.

The HAD-perturbed simulations under 1.5°C and 2°C warming give a stronger near-surface heating and show soil moisture drying from the start of the growing season onwards. This is a feature of the climatological mean response under 2°C warming (solid red line in Fig. 5a, and see Sect. 4.1), but it is more pronounced under the circulation of 2018. In contrast to 370 the EC- and MPI-perturbed simulations and the HAD-perturbed simulations under 3°C warming, precipitation decreases in AMJ. Under 3°C warming, the response in the hydrological budget terms is fairly similar to the EC-perturbed simulations, yet the response in the near-surface temperature is stronger.

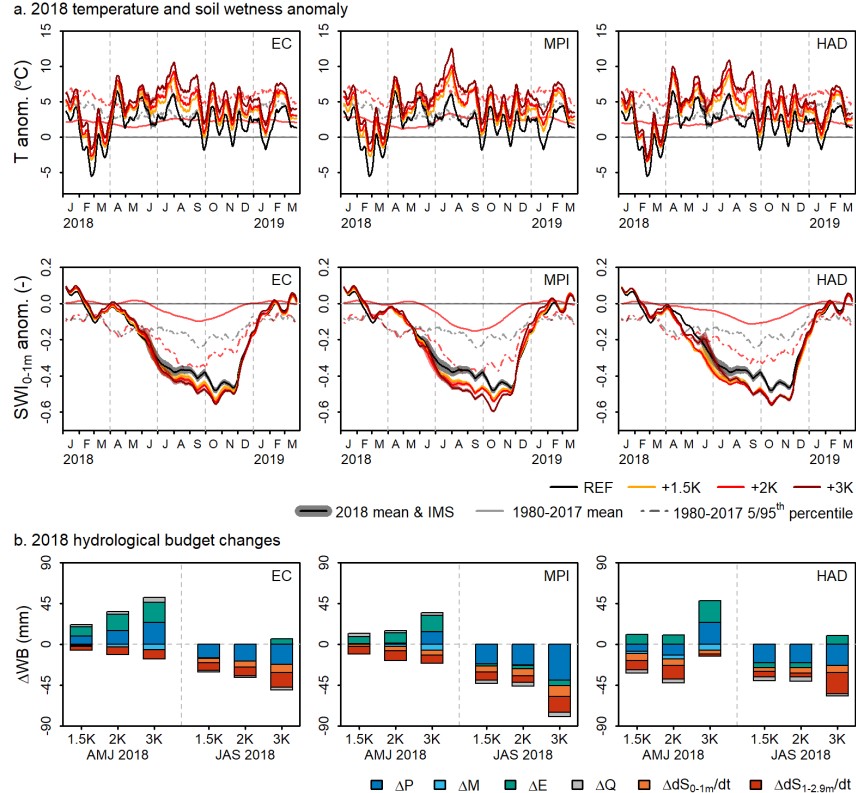

**Figure 5: Impact of global warming on the 2018 near-surface temperature, soil wetness and the hydrological budget for 1.5°C,**
**2°C and 3°C global warming, for the EC- (left column), MPI- (middle) and HAD- (left) perturbed simulations. a) Time series of the basin-mean anomaly in the 2018 near-surface temperature and the top 1m SWI with respect to the present-day climatology for present-day (grey), 1.5°C (orange), 2°C (red) and 3°C (brown) global warming. Shading indicates the inter-member spread (IMS) as in Fig. 1b. For reference, the mean climate response (red solid) and 5[th] percentile for present-day (gray dashed) and +2°C (red dashed) are shown as well. b) Change in the hydrological budget over AMJ and JAS 2018 in response to 1.5°C, 2°C**
**and 3°C global warming. The hydrological budget is given by $dS_{0-1m}/dt + dS_{1-2.9m}/dt = P + M - E - Q$, with $dS_{0-1m}/dt$ and $dS_{1-2.9m}/dt$ the seasonal change in soil moisture storage in respectively the top 1 m of the soil and bottom soil layer, and the seasonally integrated fluxes $P$ = precipitation, $M$ = snowmelt, $E$ = evaporation and $Q$ = runoff. Note that $dS/dt$ is negative over AMJ and JAS (soil moisture depletion), so that a negative response $\Delta dS/dt$ corresponds to an increase in soil moisture depletion under global warming, as can be seen in a) bottom row.**



# 5    Impact on drought and heat

## 5.1    Drought severity and frequency

To further quantify the impact of global warming on drought occurrences and severity in the west-central European river basins under 2°C warming, we determine the basin-mean drought deficit volume, duration and intensity for all years in the 1980-2020 period under present-day and +2°C conditions (Fig. 6). Under present-day conditions, the severity of the 2018 drought episode clearly stands out in both duration and intensity. Next in line is the 2003 drought episode, which has comparable duration but smaller mean intensity than 2018, and is indeed known for its severe hot and dry conditions and associated societal and economic impacts in central Europe (e.g. Rebetez et al., 2006, Fischer et al., 2007). Furthermore, the 2011 spring drought (Trachte et al., 2012) and the 2020 drought (Bissolli, 2021, Rakovec et al., 2022) are noteworthy. The 2019 soil moisture drought severity (Bissolli, 2020, Rakovec et al., 2022) is in reality likely more similar to the 2020 drought event. Comparison with observations (EOBS) shows an underestimation of the simulated precipitation deficit in the 2019 growing season, while in the 2020 growing season the precipitation deficit is overestimated in most members and the temperature anomaly is somewhat higher than observed.

Under 2°C warming the drought frequency strongly increases compared to present-day conditions, reflecting the on average drier soil moisture conditions in summer and autumn. However, the drought response is highly non-linear and several drought episodes emerge that exceed the historic 2003 drought severity. 2018 is still the most severe drought in the 1980-2020 period under 2°C warming, but the deviation from other years decreases; 1983, 1989 and 2020 are more similar to 2003 under 2°C warming. The frequency of droughts exceeding the present-day 2003 episode more than doubles, occurring on average once every eight years. This is found for all PGW simulations, irrespective of the GCM supplying the perturbations, although the response in drought intensity in the PGW simulations based on +2K|MPI is generally stronger than for the other two GCMs.

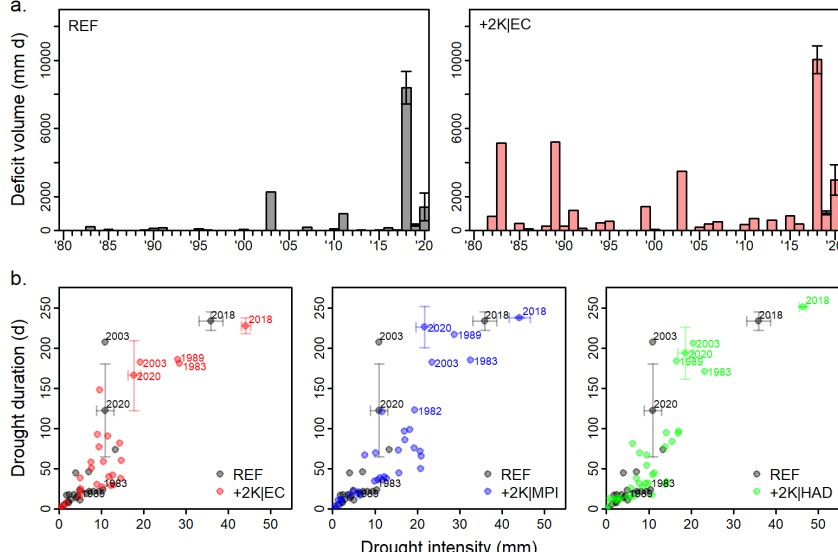

**Figure 6: Impact of 2°C warming on drought severity in 1980-2020, expressed as a) the soil moisture deficit volume and b) the drought duration and intensity. Shown is the annual maximum drought per hydrological year (April - March) under reference (grey) and +2K conditions for the EC-perturbed simulations (red) in a), and for reference (grey) and each of the PGW-simulations EC (red), MPI (blue) and HAD (green) in b). Error bars show the ensemble mean +/- 1 standard deviation for 2018-2020. The inter-member spread in the 2020 deficit volume originates mainly from spread in the drought duration. For some members the 2020 drought is split in two consecutive drought episodes, see Fig. 7a for the ensemble mean drought evolution.**



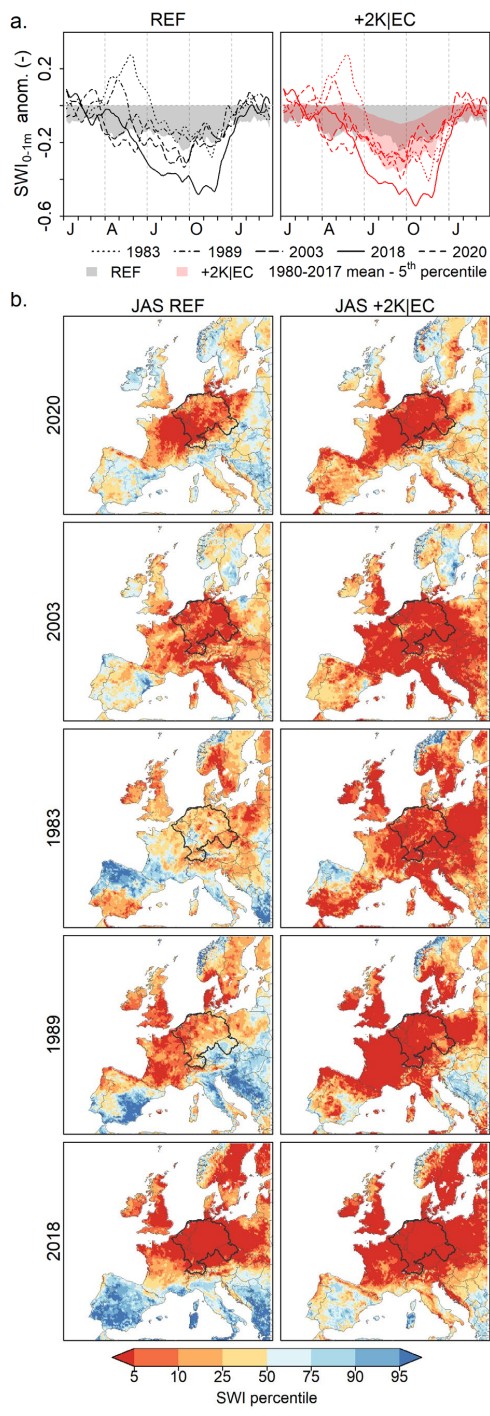

Figure 7: Present-day and future analogues of the top 5 future droughts, all exceeding the severity of the present-day 2003 drought in the west-central European river basins under +2K|EC. a) Annual cycle of the SWI anomaly for present-day (left) and +2°C conditions (right) as in Fig. 5a, but with the 1980-2017 mean-5th percentile envelope shaded. b) Spatial structure of the top 5 droughts in JAS, showing the JAS-mean SWI percentile with respect to the 1980-2017 period (climREF). Droughts are shown in order of increasing basin-mean drought severity under 2°C warming. Dark red colors indicate severe drought conditions.





For 2018, the drought onset and ending under global warming occur at roughly the same time as for the present-day event in +2K|EC for all warming levels, with only small differences between the individual members (Fig. 5a). The drought duration is thus hardly affected compared to REF, while the drought intensity shows a 23% increase (44 mm in +2K|EC compared to 36 mm in REF), resulting in a 20% increase in the drought deficit volume under 2°C warming (Fig. 6). The

drought onset in +2K|MPI and +2K|HAD occurs somewhat earlier than in +2K|EC, and the increase in drought deficit volume is stronger (+25%, resp. +39% under 2°C warming). Under 1.5°C global warming the increase in 2018 drought deficit volume is slightly smaller, while a larger intensity and deficit volume are simulated under 3°C warming (EC and MPI) with the tendency to shorter drought episodes owing to springtime precipitation increases (all simulations). Tab. C1 summarizes the findings for all warming levels and GCMs.

The increase in drought severity is surprisingly strong for the years 1983 and 1989. Under present-day conditions, 1983 and 1989 were not marked as severe drought periods in most part of the study area (see Fig. 7 where we present the basin-mean soil moisture evolution and spatial drought structure in JAS for the top five future droughts for present-day and 2°C warming). Yet, under the specific circulation conditions in 1983 and 1989, the globally warmer climate background results in strongly reduced precipitation, increased evaporation (1983 only) and soil drying in spring and early summer, and a very

strong response in incoming solar radiation, a negative response in evaporation and very strong increases in the sensible heat flux and near-surface temperatures later in summer (see also next section). The outliers in Fig. 3 correspond to these years.

The large spatial extent of all future drought analogues is remarkable (Fig. 7b). A much larger part of Europe is affected than under present-day conditions. The drought expansion is not limited to southern Europe where climatological soil moisture drying is largest. The 2018 event, for instance, spreads in all directions, now also covering southern Sweden, Poland and the

Baltic States.

### 5.2    Co-occurring dry and hot conditions

As we have seen for the present-day 2018 drought event, the extremely dry conditions co-occur with extremely high temperatures, and while the soil moisture response to 1.5°C, 2°C and 3°C global warming is fairly modest, the local temperature response is amplified compared to the mean climate response, especially in JAS (Fig. 3). The co-occurrence of

the JAS basin-mean SWI and near-surface temperature under present-day and +2°C conditions for all years in the 1980-2020 period is shown in Fig. 8 for +2K|EC, along with the co-occurrence of the response in these variables. The present-day $T$-$SWI$ distribution generally shifts towards warmer and drier conditions under 2°C warming, with larger inter-annual variability in both variables (Fig. 8a). Strong responses in soil moisture drying (1983,1989) co-occur with particularly strong temperature increases, but also some years with small soil moisture responses (2003, 2018) exhibit fairly strong warming

(Fig. 8b), which contributes to an increase in inter-annual variability in both temperature and soil wetness. The increase in inter-annal variability is somewhat less pronounced in the HAD-perturbed simulations, see Fig. C8.

The non-linearity in the drying and warming response is related to the transition of predominantly energy-limited to more moisture-limited evaporation in the west-central-Europe (Schär et al., 2004, Lenderink et al., 2007, Seneviratne et al., 2010, Zscheischler and Seneviratne, 2017). Years with weakly moisture-limited evaporation under present-day conditions

(showing relatively small differences between actual and potential evaporation; see Fig. 1b, $E$ and $E_p$) may shift to strongly soil-moisture-limited energy balance regimes under PGW, through a decrease in precipitation and enhanced early season evaporation. Conversely, several years with present-day JAS temperature and soil wetness comparable to 1989 and 1983 show a much weaker response. The specific large-scale circulation (variability) and in particular the corresponding precipitation response are important factors in the initiation of soil moisture drying and amplified warming. However, the

drought evolution is unique for each year and disentangling the exact drivers of the amplified drying response is outside the




scope of this paper.

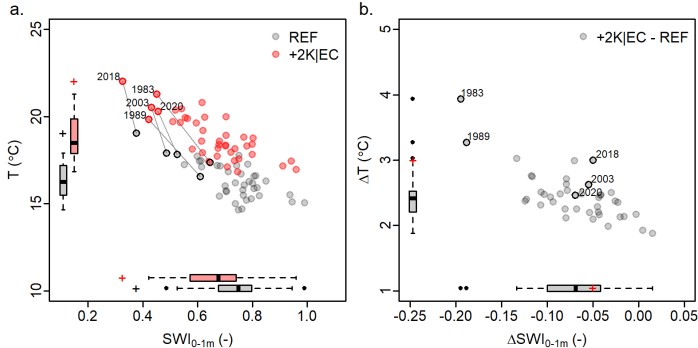

**Figure 8: Scatterplot of the 1980-2020 JAS basin-mean near-surface temperature and SWI response. a) Absolute values under present-day (grey) and +2°C (red) conditions and their distribution (boxplots). b) Response to 2°C warming and its distribution (boxplots). Results are based on the EC-perturbed simulations. 2018 is indicated with + in the boxplots. The top five driest years under 2°C warming are marked.**

## 6    Discussion

We have examined the implications of global warming for future droughts in west-central Europe, by employing PGW experiments for the 1980-2020 period. PGW experiments have previously been used – from very simple uniform warming experiments to more advanced perturbations – to examine changes in heavy precipitation (e.g. Attema et al. 2014, Prein et al. 2017, Lenderink et al. 2019), disentangle the contribution of different drivers to amplified Mediterranean warming and drying (Kröner et al. 2017, Brogli et al. 2019) and provide future weather scenarios of extreme precipitation events (Klein Tank et al. 2014, Lenderink and Attema, 2015). The simulations performed in this study allow for a systematic examination of the impact of global warming on droughts by comparing future drought analogues with present-day events, and the simulations provide anecdotical examples of the impact of global warming to complement conventional approaches based on large-ensemble climate simulations. Where the conventional approaches generate probabilistic estimates of changes in climate events, the PGW approach generates storylines of plausible future climate events that we can relate to. Storylines make future climate risks more tangible and better communicable than statistics (Hazeleger et al. 2015, Shepherd et al. 2018).

In the following we compare our results with studies based on large-ensemble simulations and discuss the implications of our findings.

### 6.1    The future 2018 drought: drier, hotter and bigger

It is generally hypothesized that under globally warmer conditions droughts set in earlier, last longer and are more intense, if conditions favoring a drought occur (Trenberth et al., 2014, Seneviratne et al., 2010). Under the anomalously persistent atmospheric blocking conditions of 2018, global warming leads to increases in precipitation in early spring that (partially) compensate the increase in evaporation, limiting an earlier drought onset, while precipitation increases in autumn terminate the 2018 drought episode at the same time as under present-day conditions. However, the drought intensity indeed increases, yielding a 20% (EC) to 39% (HAD) increase in drought severity under 2°C global warming. From an impact perspective, this is a considerable increase, with substantial costs to society and nature already under present-day conditions (Van Hussen et al., 2019, De Brito et al., 2020, Toreti et al. 2019, Schuldt et al., 2020, Beillouin et al., 2020, Senf and Seidl, 2021). The increase in drought severity in summer co-occurs with an increase in local summer temperature that is considerably larger than the mean climate response. The combination of increasing heat and drought leads to even stronger increases in stress




on nature and society, and may enhance tree mortality (Allen et al., 2010), wild fire risk (Krikken et al. 2021), crop yield losses (Matiu et al., 2017) and water quality deterioration (Wolff and Van Vliet, 2021, Van Vliet et al., 2011) impacting ecosystems, industry, and energy and drinking water production. Moreover, the increase in drought extent under global

warming, which also emerges in conventional ensemble simulations (e.g. Hari et al., 2020 and Samaniego et al., 2019), implies that much larger parts of Europe will be affected simultaneously.

While the 2018 soil moisture response is considerable in absolute sense, the soil moisture drying is small compared to the climatological response to 2°C global warming. The same applies for 2003, the second largest drought episode in the 1980-2020 simulation period. While this has the physical explanation that the soil moisture response in drought years is limited

by the strongly moisture-constrained conditions, this behavior may in part be explained as a PGW-artifact. Extreme climate events occur when extreme drivers compound. In order for an event to become more extreme under PGW, all, or at least most, drivers of the event must be 'pushed' towards a more extreme state by the perturbations. This is very likely for temperature under the strong temperature perturbation, yet it is not so obvious for e.g. atmospheric stability and wind direction. Since there is a larger number of pathways for any other year to become more extreme, also statistically it is more

likely that years in which moderately dry present-day conditions prevail show a much stronger drying response than the extreme 2018 drought, and become more similar to the 2018-event under PGW.

More extreme drought occurrences than the 2018+2K event are plausible in a globally warmer world, in particular through an increase in drought duration driven by even more persistent or longer sequences of atmospheric blocking conditions, drier antecedent winter and spring conditions and/or stronger climate induced spring precipitation decreases than derived from

the PGW experiments. Van der Wiel et al. (2021) also follow a storyline approach, but sample drought events from a very large ensemble of transient global climate model simulations (EC-EARTH) that match or exceed the 2018 drought conditions in the Rhine basin under present-day and globally warmer conditions. They indeed find a set of events with a slightly stronger drying response in spring than under PGW with EC-perturbations, but results are similar to the HAD-perturbations. This is an elegant approach to find future analogues of present-day events as well, but such approach relies on a very large

GCM(RCM) ensemble, and the atmospheric circulation of the future analogues doesn't necessarily match the present-day circulation so a one-to-one comparison of present-day and future events is not possible.

### 6.2    The future of historic summers: moderately dry summers respond more strongly than extremes

The climatological mean soil moisture drying response under PGW closely resembles results based on ensembles of transient climate model simulations. However, soils are generally replenished to present-day (near-saturation) soil moisture levels in

winter under PGW, whereas e.g. Ruosteenoja et al. (2018) and Van der Linden et al. (2019) find a small drying response throughout winter in transient simulations. This discrepancy could be explained by differences in GCM/RCM structure and model resolution. Also, the absence of high-frequency changes in the large-scale atmospheric circulation under PGW can explain these differences. Brogli et al. (2019) compare the full climate change response in transient simulations with the response under PGW and show that the high-frequency changes contribute to an increase in evaporative demand and decrease

in precipitation in west-central European summer and to a reduction of the mean precipitation increase in winter. The 'error' we make by neglecting these changes is likely small (De Vries et al. 2022), but may lead to a slight underestimation of mean soil moisture drying and moderate drought occurrences.

The increase in severe drought occurrences under PGW is in the range of changes in the drought intensity-frequency-distribution derived from transient climate model simulations, with a doubling of 2003-like soil moisture droughts under

3°C warming found by Samaniego et al. (2018), and 2018-like drought conditions (SPEI) becoming the new normal within the second half of the 21st century according to Toreti et al. (2019). The increase under PGW occurs under the historic large-



scale atmospheric circulation, i.e. independently of changes in the frequency of atmospheric blocking conditions, and is owing to particularly strong soil moisture and temperature responses in years with moderately dry present-day conditions. While the transition from energy-limited to moisture-limited evaporation regimes can explain the co-occurrence of strong

temperature increases and soil moisture drying, the relative contribution of different mechanisms that cause these strong responses vary widely and obscure a general picture. The results are robust with respect to the selected GCM to derive the perturbations.

The increase in frequency of extreme drought occurrences implies shorter recovery times between events, amplifying the impacts (Zscheischler et al. 2020). In particular ecosystems can exhibit increased vulnerability to a second compared to an

initial drought (Anderegg et al., 2020, Bastos et al., 2021). Temporally compounding financial losses may affect for example the agricultural sector and industry with supply chains that depend on inland shipping.

### 6.3    Climate adaptation studies

The exploration of future analogues of historic (extreme) events is useful for different applications addressing climate process understanding, impact assessment or stress testing of climate adaptation strategies (Shepherd et al, 2018, Sillmann

et al, 2021). The PGW-simulations presented here have been used to investigate the hydrological impact of land use change and ecosystem adaptation to climate change, by forcing a hydrological model with time-variant vegetation parameters with the PGW simulations (Bouaziz et al., 2022). Bouaziz et al. show that increases in rooting depth in response to climate change result in enhanced evaporation and decreases in river runoff. It would be interesting to examine the impact of the hydrological changes on the meteorological and soil moisture drought development, which is in principle feasible in the PGW setup.

### 7    Conclusions

Droughts and associated heat waves form a threat to society and nature, as demonstrated in recent years in west-central Europe, and presently again by the 2022 drought which affects large parts of Europe. To develop adaptation strategies, information of changes in drought risk under ongoing global warming is required. In this study we have examined the implications of global warming for future drought severity in west-central Europe, by systematically perturbing the 1980-

2020 period towards future climate conditions using the pseudo global warming (PGW) approach. The reference experiment has been carried out with the RCM KNMI-RACMO2 forced by large-scale information from the ERA5 reanalysis. The PGW-experiments use monthly mean changes in temperature, humidity and winds derived from GCM projections. In this approach, the signal-to-noise ratio of the climate response is optimized and changes in droughts can be directly related to events and their societal impact in the recent history. Therewith the experiments provide tangible examples of what global

warming entails and may serve as a tool to examine and communicate adaptation strategies.

Under 2°C warming almost all years in the 1980-2020 period show a decrease in soil moisture availability in (spring), summer and autumn, consistent with results based on transient climate model simulations. Under the circulation of 2018 the temperature response is strongly amplified, while the soil moisture response is limited by the strong moisture-constrained evaporation during present-day conditions. Nevertheless, the soil moisture deficit volume increases by 20% to 39% under

2°C global warming, depending on the perturbing GCM, owing to an increase in drought intensity. The drought duration is barely impacted owing to increasing precipitation in spring, autumn and winter.

We furthermore show that the response in soil moisture drying and temperature can be particularly large for years with moderately dry conditions in the present-day climate. This implies that years that went hardly noticed in the present-day climate may emerge as very dry and hot years in a warmer world. Using present-day thresholds, the drought frequency





strongly increases under 2°C warming, with more severe than 2003-like deficit volumes occurring every eight years, and exhibiting strongly enhanced temperatures. This shows that even without taking into account changes in the frequency of atmospheric blocking conditions, the drought risk in west-central Europe is strongly enhanced by the drought intensification and increase in frequency, yielding shorter recovery time between events for nature and society.

**Acknowledgements**

The authors acknowledge the E-OBS dataset from the EU-FP6 project UERRA (http://www.uerra.eu) and the Copernicus Climate Change Service, and the data providers in the ECA&D project (https://www.ecad.eu). This research was supported by the European Union Horizon 2020 IMPREX (www.imprex.eu, Grant agreement No 641811) and EUCP projects (https://www.eucp-project.eu, Grant agreement No 776613).

**Code/Data availability**

Model data are available upon request directed at the corresponding author.

**Author contribution**

All authors were involved in the study design. EEA carried out the analyses and prepared the manuscript. Results were discussed with all co-authors and all co-authors commented extensively on early drafts. EvM performed all RCM simulations and prepared the perturbations together with HdV.

**Competing interests**

The authors declare that they have no conflict of interest.

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



# Appendices

## A Simulations

**Table A1: Perturbation and greenhouse gas forcing periods for the PGW simulations. The warmingperiod is the period in which the target warming level is reached in the GCM simulations, and is used to determine the perturbations. The PGW simulations are forced with projected aerosol and greenhousegas concentrations for the years shown under GHG.**

| Simulation name | period | EC-EARTH v2.3 16 members | | HadGEM2-ES r1-r4 | | MPI-ES-LR r1-r3 | |
| --- | --- | --- | --- | --- | --- | --- | --- |
| | | warming period | GHG | warming period | GHG | warming period | GHG |
| 2018+1.5K | 2018-2020 | 2037-2066 | 2058-2060 | 2027-2056 | 2048-2050 | 2036-2065 | 2057-2059 |
| 2018+2K | 2018-2020 | 2048-2077 | 2069-2071 | 2036-2065 | 2057-2059 | 2048-2077 | 2069-2071 |
| clim+2K | 1979-2017 | | 2030-2068 | | 2018-2056 | | 2030-2068 |
| 2018+3K | 2018-2020 | 2069-2098 | 2090-2092 | 2053-2082 | 2074-2076 | 2068-2097 | 2089-2090 |

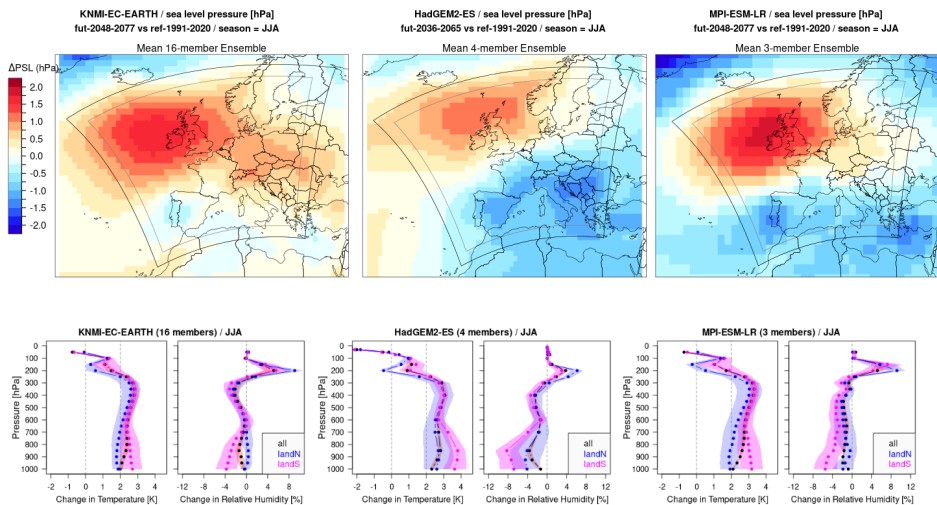

**Figure A.1: Top row: Surface pressure perturbations corresponding to 2K global warming derived from EC-EARTH (left), HadGEM2 (centre), and MPI (right) for JJA. Bottom row: vertical profiles of temperature and relative humidity derived from the same GCMs. "All" refers to all grid cells enveloped by the solid line indicating the edges of the RCM-domain; "landN" refer to land points north of 50 ºN; "landS" to cells in between 35 and 50 ºN. The shading indicates the spread across the regions.**





**B Atmospheric evaporative demand**

The computation of soil evaporation and transpiration in HTessel both use a resistance approach (ECMWF, 2009), see Eq. B1.

$$E = \frac{\rho_a}{r_a + r_i}[q_L - q_{sat}(T_{skin})] \tag{B1}$$

where $r_i$ is the surface resistance, $r_a$ the aerodynamic resistance, $\rho_a$ is the air density, $q_L$ the specific humidity of the lowest atmospheric model level and $q_{sat}$ is the saturated specific humidity at skin temperature ($T_{skin}$). $r_i$ is replaced by a canopy resistance ($r_c$) for transpiration and by a soil resistance ($r_{soil}$) for soil evaporation. $r_c$ is modeled following Jarvis (1976), and is a function of the minimum stomatal resistance ($r_{S,min}$), the leaf area index ($LAI$), the downward short-wave radiation ($R_s$), unfrozen root soil water ($f_{liq\theta}$) and atmospheric water vapour deficit ($D_a$) (Eq. B2). The $r_{soil}$ is a function of a minimum soil resistance ($r_{soil,min}$) and unfrozen soil water content in the top layer ($f_{liq,\theta_1}$). (Eq. B3).

$$r_c = \frac{r_{S,min}}{LAI} \cdot f_1(R_s) \cdot f_2(f_{liq,\theta}) \cdot f_3(D_a) \tag{B2}$$

$$r_{soil} = r_{soil,min} \cdot f_2(f_{liq,\theta_1}) \tag{B3}$$

$f_1$, $f_2$ and $f_3$ are 1 for unconstrained conditions and larger than 1 for constrained conditions (see ECMWF, 2009). In order to determine the potential evaporation, $f_2$ for soil evaporation and transpiration and $f_3$ for transpiration are set to 1, while $f_1$ and all other variables (e.g. temperature, humidity) are taken from the prognostic computation with actual evaporation.





## C. Response

### C.1 Climate response

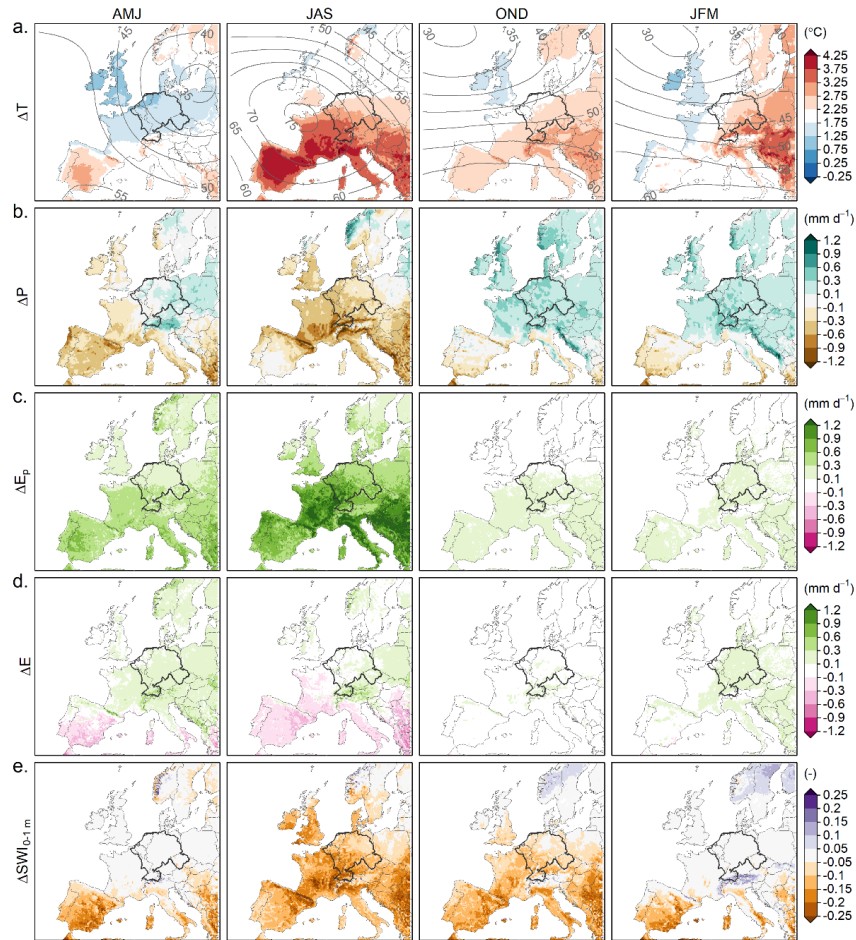

**Figure C1: As Figure 2, but for clim+2K|MPI**



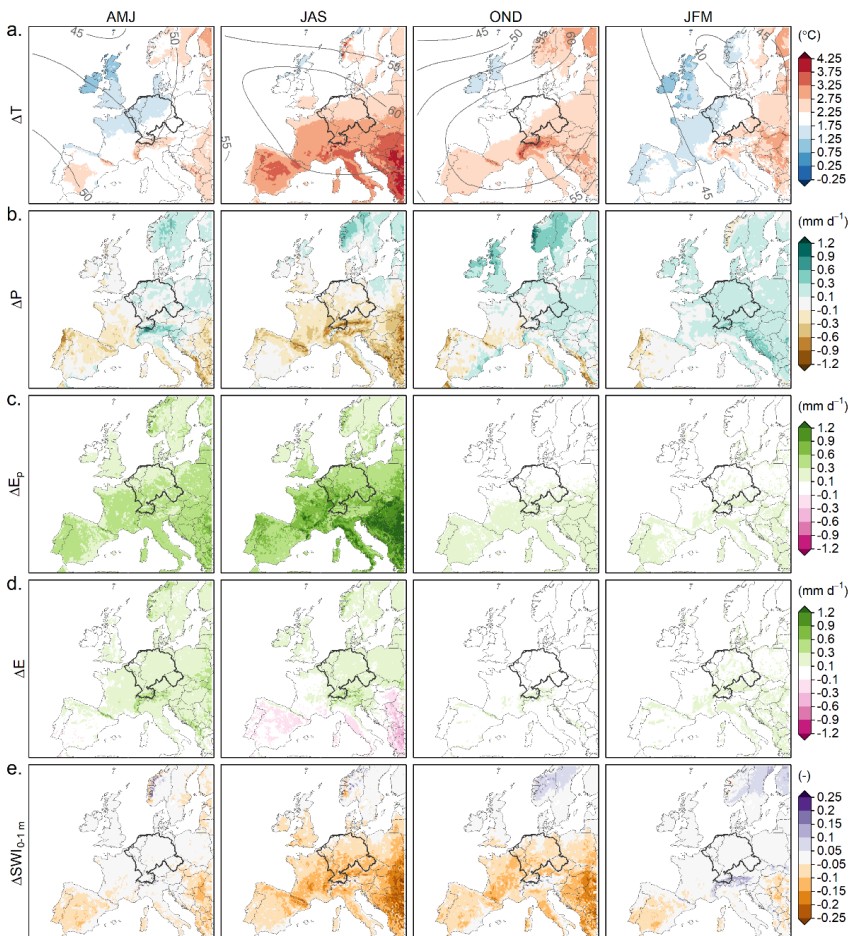

Figure C2: As Fig. 2, but for clim+2K|Had



## C.2  2018 response



**Figure C3: As Fig. 4, but for clim+2K|MPI**

**Figure C4: As Fig. 3, but for clim+2K|MPI**

**Figure C5: As Fig. 4, but for clim+2K|HAD**

**Figure C6: As Fig. 3 but for clim+2K|HAD**





### C.3 Impact on drought severity and frequency

**Table C1: Drought severity of the 2018 drought for present-day conditions (REF) and for 1.5K, 2K and 3K global warming. Listed are the ensemble mean and (standard deviation).**

|       |        | Duration (d) | | Intensity (mm) | | Deficit volume (mm d) | |
|-------|--------|------|------|------|-----|-------|-------|
|       | REF    | 233  | (11) | 36   | (3) | 8392  | (968) |
|       | +1.5K  | 229  | (12) | 42   | (2) | 9562  | (810) |
| EC    | +2K    | 228  | (10) | 44   | (1) | 10049 | (548) |
|       | +3K    | 221  | (11) | 46   | (1) | 10142 | (614) |
|       | +1.5K  | 238  | (1)  | 42   | (1) | 10085 | (359) |
| MPI   | +2K    | 238  | (1)  | 44   | (3) | 10497 | (621) |
|       | +3K    | 235  | (1)  | 47   | (2) | 11053 | (538) |
|       | +1.5K  | 252  | (1)  | 46   | (2) | 11493 | (606) |
| HAD   | +2K    | 251  | (1)  | 46   | (1) | 11646 | (222) |
|       | +3K    | 234  | (20) | 45   | (3) | 10411 | (770) |

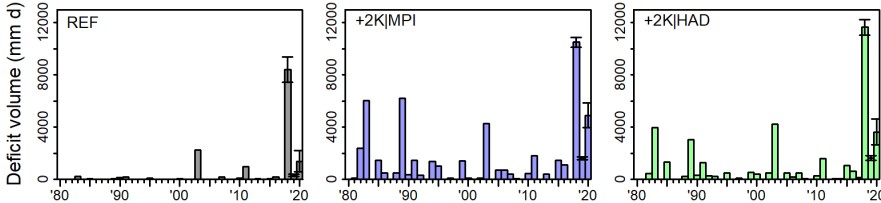

**Figure C7: As Fig. 6a, but for the MPI- and HAD-perturbed simulations.**

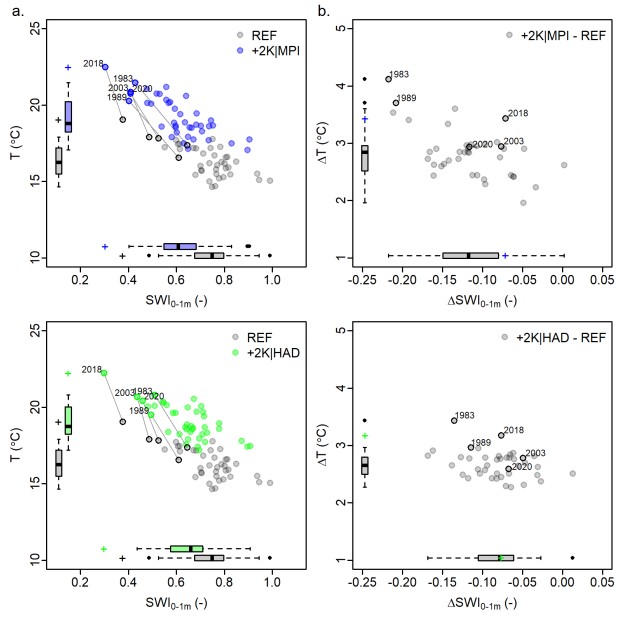

**Figure C8: As Fig. 8, but for the MPI- and HAD-perturbed simulations.**