# Peer review of "The 2018 west-central European drought projected in a warmer climate: how much drier can it get?"

_EGUsphere, 2022_

## Referee Comment (RC2)

**General remarks**

The paper is a comprehensive study of the 2018 west-central European drought focussing on soil wetness deprivation, and what that drought would have been like in a world with different levels of climate change using the pseudo global warming method and analogues. The study is technically rich, as the analysis is done with different global climate models for input to the regional model providing a cross-check for the results, as well as content-wise rich with comparison to other drought events and a climatological test of droughts in this region and elaborate placement of their findings in existing literature.

I find this research very relevant as it produces new insights in the attribution of European droughts. The analysis of the drought is well executed and technically sound, including temperature, precipitation and evaporative demand. Limitations to the analysis are also reflected on accurately. The paper is lengthy, though, and could benefit from a reduction in size (I have tried to provide some small suggestions for this in the minor comments below). The English language is of high quality and the paper reads well.

**Detailed remarks**

You have used "(not shown)" six times in this paper (L221, L235, L276, L279, L329, L340). This is too often and becomes a hindrance since it implies a demand of trust. Either show what you claim, rephrase, reference to a paper that proofs a comparable outcome to the one you found, or delete the claim. Please try to reduce the "(not shown)" to one or two times. Since you have a large appendix already, you might consider placing some of these aspects in the supplementary material to prevent the paper from becoming larger.

Not all graphs are easily readable. I reviewed using a print-out and could not read Figure 1 and Figure 5a, and to some extent Figure 3. I have put a more detailed explanation below, but would encourage giving figures a check-over for readability.

**Minor comments**

| | |
|---|---|
| L52-L56 | This is 1 very long sentence. Please shorten for ease of understanding. |
| L67-L69 | This sentence could use a reference to back up the claims. For the Mediterranean precipitation research, you could cite Zappa et al., 2017 (http://dx.doi.org/10.1175/JCLI-D-16-0807.s1). |
| L99-L100 | Is your study trying to provide something like the 'common framework' published by Shepherd et al., 2016? (http://dx.doi.org/10.1007/s40641-016-0033-y) |
| L132-134 | You are altering a large number of variables to create the counterfactual worlds. Could you argue why it is acceptable to meddle with the model to such a large extent without loosing physical self-consistency? In essence, you are altering the consequences of climate change (T, P, SH, etc.) instead of (only) the causes of climate change (GHG, SST, etc.). I am aware that the method requires this to create a counterfactual world, but I would like to hear the validation in 1 or 2 short sentences (could be in discussion you find that a better fit). |
| L145-146 | Please add the resolutions of the individual EC-EARTH v2.3, HadGEM2-ES and MPI-ESM-LR GCM's. |
| L169-173 | The reader might benefit from adding an equation to show this second step as well. |
| Figure 1 | The b panels are not readable at all, please change colours and either make the graph bigger or lines a little thinner. Ep is not mentioned in the caption. The legend in the b graphs is incomplete (shading is not mentioned, for instance). |
| L203 | I do not believe the pressure anomalies are shown? Or are they the once in the supplementary material? Please refer if that's the case or show what you are claiming or reference to a paper that shows it. |

| | |
|---|---|
| L261 | The reader could benefit from adding which colour line to look at when referring to Figure 3a. Also, to add which colour to look at with "mean response is rather large…" |
| L266 | "increasing cloud cover…" is not shown anywhere, or do you mean an interpretation of solar radiation in Figure 3b? Please clarify. |
| L282-L283 | Figure C1 and C2 show vastly different GPH z500 patterns. Should this be mentioned or explained? Is this of significance for the analysis? |
| Figure 3 | The legend is incomplete, please also mention what the shading etc. stands for. The caption is a bit of an essay. I would suggest placing the method part that right now is at the end of the caption, including the equation, to the main text or supplementary material. From a print the graph is difficult to read, which is a pity since it shows some essential and interesting things. You could either make it bigger or reduce the thickness of the median (black or red). |
| Figure 5a | The lines are almost all the same colour, and to thick to interpret. There are even two lines that are both red. Please update the graph, make the lines thinner or the graph bigger, and choose colours that are further apart from each other. The caption is again a bit of an essay with a method at the end that could be explained in the main text or in supplementary material. |
| L395 | Just to make sure, is the comparison with E-OBS done by the paper you are citing in the previous sentence? I was trying to find a graph that shows the comparison, but if it is in that paper, please clarify this in the sentence ("the authors found…" or something like that). |
| L490-496 | I do not think you need to state the obvious, plus you are not testing these impacts in this paper. You could save space by deleting this section. |
| L510-L516 | *(optional)* You could add spectrally nudged storylines to the list of options, it will allow for drought intensification studies, but not changes in dynamics (which you claim in L571-573 is plenty) with the benefit of a very small size ensemble (van Garderen & Mindlin 2022, https://doi.org/10.1002/wea.4185). However, since I am the author and it is for a region outside of Europe, feel free to ignore this comment. |
| L521-525 | Could it be that the discrepancy has anything to do with altering symptoms of climate change and not causes? See also my comment for L132-L134 |
| L543-545 | In the absolute sense the referencing is correct, since the papers do mention analogues as well. However, storylines and analogues are not the same thing, and the emphasis of the paper cited is on storylines. Perhaps succinctly place the analogues in the context of storylines without doing another literature review (which you have already done). |

The comments I made are minor, and I am looking forward to seeing this paper published.
Best,
Linda van Garderen

---

## Author Comment (AC1)

**Reply to Patrick Ludwig**

https://doi.org/10.5194/egusphere-2022-954-RC1

We thank Patrick Ludwig for his positive assessment of our manuscript and appreciate the suggested improvements. Please find our responses to the comments below in blue.

Review of egusphere-2022-954 '*The 2018 west-central European drought projected in a warmer climate: how much drier can it get?*' by Aalbers et al.

**General comments:**

In their paper `*The 2018 west-central European drought projected in a warmer climate: how much drier can it get?*´, the authors investigate the impact of global warming on soil moisture drought severity for the year 2018 in west-central Europe. With this aim, pseudo-global warming (PGW) experiments with a regional model were performed, forced by three GCMs for three different global warming levels (+1.5K, +2K, +3K). Their results show, that under global warming the 2018 drought episode experiences strongly enhanced summer temperatures, but a fairly modest soil moisture drying response compared to the change in climatology as evaporation is already strongly moisture-constrained during present-day condition. In more general, the authors show that the drought risk in west-central Europe is strongly enhanced under global warming.

This work provides is a valuable contribution to our understanding of the consequences of global warming on extreme events (here drought) observed so far. The methods, techniques and the experimental setup applied in this study are sound and state of the art; the use of different GCMs (considering an ensemble mean for each GCM) to create the perturbations under global warming provides robust estimates of how global warming might modify the 2018 drought event. Beside some minor comments, this study is a very valuable contribution to NHESS.

**Minor Comments**

L62: What does the abbreviation SPEI stand for?

SPEI is the Standardized Precipitation - Evaporation Index. We will clarify this in the revised manuscript.

L135ff: Usually, global warming levels (GWL) are defined based on the pre-Industrial reference period (1850-1900). In the IPCC AR6 report, the global warming between 1850-1900 and 2011-2020 is estimate with 1.09K, thus meaning roughly GWL1. Based on the Paris agreement, the long-term temperature goal is to keep the rise in mean global temperature to well below 2K (GWL2) above pre-industrial levels, and preferably limit the increase to 1.5K (GWL1.5). To avoid any confusion, between the IPCC based and your GWLs (based on the 1991-2020 period), a short note might be helpful.

We will add a sentence to emphasize this in the revised manuscript.

L144ff: Can you comment briefly about the model quality of the 3 GCMs in comparison with observation? Is for example the temperature bias of the reference period you use (1991-2020) the same for all models? Or are the GCMs that already show enhanced global warming for this period in comparison with the observational record?

We will. Note however that the impact of model biases in the reference period in the GCM is limited to the impact on the perturbations, since the RCM reference run is driven by ERA5.

L258: Should read '*over the British Isles*'

Thanks, we will change this.

L393: The 2011 spring drought is not indicated in Figure 6b. Could you add this year to the graphs?

Yes, we will add the 2011-label to the plots.

Figure1: $E_p$ is missing in the list of variables in (a); the orange lines for the observations (obs) for 2018-2019 are hard to see; consider to use a more striking color.

Thanks for noting, we will add $E_p$ and do our best to find a better color.

Figure2: Consider moving the column with JFM as the first column; would better reflect the course of the year.

We chose to start with April as the start of a hydrological year, but we don't have a strong preference and can indeed show JFM in the first column.

---

## Author Comment (AC3)

**Reply to Linda van Garderen**

https://doi.org/10.5194/egusphere-2022-954-RC2

We thank Linda van Garderen for her thorough reading, constructive comments and the overall positive assessment of our manuscript. In the following, we address her comments point-by-point.

General remarks

The paper is a comprehensive study of the 2018 west-central European drought focussing on soil wetness deprivation, and what that drought would have been like in a world with different levels of climate change using the pseudo global warming method and analogues. The study is technically rich, as the analysis is done with different global climate models for input to the regional model providing a cross-check for the results, as well as content-wise rich with comparison to other drought events and a climatological test of droughts in this region and elaborate placement of their findings in existing literature.

I find this research very relevant as it produces new insights in the attribution of European droughts. The analysis of the drought is well executed and technically sound, including temperature, precipitation and evaporative demand. Limitations to the analysis are also reflected on accurately. The paper is lengthy, though, and could benefit from a reduction in size (I have tried to provide some small suggestions for this in the minor comments below). The English language is of high quality and the paper reads well.

Detailed remarks

You have used "(not shown)" six times in this paper (L221, L235, L276, L279, L329, L340). This is too often and becomes a hindrance since it implies a demand of trust. Either show what you claim, rephrase, reference to a paper that proofs a comparable outcome to the one you found, or delete the claim. Please try to reduce the "(not shown)" to one or two times. Since you have a large appendix already, you might consider placing some of these aspects in the supplementary material to prevent the paper from becoming larger.

Thank you for pointing this out. We will reduce the number of *not shown*s in the revised manuscript:

L221: As a consequence, the sensible heat flux strongly increases (not shown), ..

This concerns the increase in sensible heat flux during the summer of 2018, which logically follows from the increase in evaporative demand (shown), and decrease in evaporation (shown) and soil moisture (shown). In our opinion, here no additional figure is needed.

L235: For deeper soil layers the winter precipitation is insufficient to fully replenish the soils to normal levels, and the anomalously dry conditions persist throughout 2019 (not shown).

Since the focus of this manuscript is on the top 1 m of the soil, we decided to limited the figures to variables that are directly relevant for the top 1 m of the soil. However, readers may remember that groundwater levels were lower than normal throughout winter and spring 2019. To show that this is the case in our simulations as well we'll add a figure similar to figure 1(b) in the supplementary material/appendix of the revised manuscript.

L276: This feature of the response is amplified by reduced snowmelt in spring and a larger fraction of precipitation falling as rain in autumn (not shown).

We could add a figure similar to figure 5b (water balance) for the climatological mean response to the appendix.

L279: In summer and autumn, the soil moisture availability in deeper layers and runoff decrease as well (not shown).

We can change the orientation of Fig. 3, and add a subplot with the soil moisture development in deeper layers in the main text, or we can add a figure similar to Fig. 5b (response in water balance components) for the climatological mean response to the appendix (as for L276).

L340: Percolation to deeper soil layers and runoff decrease in this period as well (not shown).

Actually this is shown later in the manuscript, in Fig. 5b. We'll add a reference in the revised manuscript.

Not all graphs are easily readable. I reviewed using a print-out and could not read Figure 1 and Figure 5a, and to some extent Figure 3. I have put a more detailed explanation below, but would encourage giving figures a check-over for readability.

We're sorry that you couldn't easily read some of the figures in the printed version of the manuscript. Apart from the color in figure 1b the other reviewers did not comment on the figures, so we trust that larger (online) versions of the figures are clear. We will provide full-width versions of the multi-panel plots in the revised manuscript.

Minor comments

L52-L56 This is 1 very long sentence. Please shorten for ease of understanding.

We'll split the sentence in the revised manuscript

L67-L69 This sentence could use a reference to back up the claims. For the Mediterranean precipitation research, you could cite Zappa et al., 2017 (http://dx.doi.org/10.1175/JCLI-D-16-0807.s1).

The response pattern over Europe described in L67-69, including the response over west-central Europe (L70-71) is shown in the references in line L71. We will move these references one line up in the revised manuscript to make this more clear.

L99-L100 Is your study trying to provide something like the 'common framework' published by Shepherd et al., 2016? (http://dx.doi.org/10.1007/s40641-016-0033-y)

*L99-100: Additionally we evaluate the position of this 2018 event in the 1980-2020 period, both for present-day and for future conditions under a single warming level.*

Shepherd (2016) discusses extreme event *attribution* to climate change, and we wouldn't suggest that we do anything like that (a 41-year simulation period is way to short for attribution). However, simulating a future weather analogue not only of the extreme 2018 event, but of all summers in the 1980-2020 period, provides context and a better understanding of the changes we find for the extreme summer of 2018.

L132-134 You are altering a large number of variables to create the counterfactual worlds. Could you argue why it is acceptable to meddle with the model to such a large extent without loosing physical self-consistency? In essence, you are altering the consequences of climate change (T, P, SH, etc.) instead of (only) the causes of climate change (GHG, SST, etc.). I am aware that the method requires

this to create a counterfactual world, but I would like to hear the validation in 1 or 2 short sentences (could be in discussion you find that a better fit).

The strength of the application of a regional climate model for this purpose is that the simulations are physically consistent in the interior of the model domain by design. The perturbations itself are derived from GCM simulations and are added to *all* state variables, again to ensure physical consistency. Moreover, the perturbations are fairly small (apart from temperature) and are smoothly varying in space and time. See also e.g. Prein et al. 2016 and Shepherd et al. 2018.

Of course, we built our PGW simulations on the large-scale circulation in the present-day period, and it may be that the frequency of e.g. blocking conditions changes in a globally warmer world. However, dynamical changes are highly uncertain, and there is no physical argument against the plausibility of the present-day circulation types under globally warmer conditions.

L145-146 Please add the resolutions of the individual EC-EARTH v2.3, HadGEM2-ES and MPI-ESM-LR GCM's.

We'll add a table with characteristics of the GCMs in appendix A.

L169-173 The reader might benefit from adding an equation to show this second step as well.

Ok, we'll add that!

Figure 1 The b panels are not readable at all, please change colours and either make the graph bigger or lines a little thinner. Ep is not mentioned in the caption. The legend in the b graphs is incomplete (shading is not mentioned, for instance).

Thanks for noting, we will add $E_p$ in the caption.

The legend in figure b is spread over the top graph (colors) and bottom graph (line thickness and shading) the make full use of the space, and the colors and shading are explained in the caption. However, apparently this is not obvious to the reader, so we will combine both legends and show the legend below the graph.

We'll pick a different color for the observations to enhance the readability.

L203 I do not believe the pressure anomalies are shown? Or are they the once in the supplementary material? Please refer if that's the case or show what you are claiming or reference to a paper that shows it.

Yes they are shown as the 500 hPa geopotential height anomaly (contours in Fig. 1). We'll refer to the figure in the revised manuscript.

L261 The reader could benefit from adding which colour line to look at when referring to Figure 3a. Also, to add which colour to look at with "mean response is rather large…"

We will include in the text that 'the grey boxes show the 1980-2017 mean and inter-annual spread'.

L266 "increasing cloud cover…" is not shown anywhere, or do you mean an interpretation of solar radiation in Figure 3b? Please clarify.

The increase in solar radiation in JJASO indeed results from decreases in cloud cover (and humidity). The response in cloud cover is not shown. The response in solar radiation and relative humidity is. We will clarify this in the manuscript.

L282-L283 Figure C1 and C2 show vastly different GPH z500 patterns. Should this be mentioned or explained? Is this of significance for the analysis?

We describe this briefly in the methods section (2.2). The GCMs indeed differ in their regional response pattern. That is the reason for using perturbations derived from three different GCMs. Although the GCM is 'recognizable' in the PGW simulations, the results are overall fairly robust with respect to the GCM that is used to derive the perturbations.

Figure 3 The legend is incomplete, please also mention what the shading etc. stands for. The caption is a bit of an essay. I would suggest placing the method part that right now is at the end of the caption, including the equation, to the main text or supplementary material. From a print the graph is difficult to read, which is a pity since it shows some essential and interesting things. You could either make it bigger or reduce the thickness of the median (black or red).

As written above, we will enlarge the figure to improve the readability in a printed version of the manuscript. We will place the legend outside the subplot and clarify the shading. Concerning the caption, we prefer to leave the explanation that is required to understand the figure within the caption, for the reader to understand the figure without having to go through the text / supplement. In order to understand the text in which this figure is discussed, the explanation provided in the caption isn't directly needed and would only distract.

Figure 5a The lines are almost all the same colour, and to thick to interpret. There are even two lines that are both red. Please update the graph, make the lines thinner or the graph bigger, and choose colours that are further apart from each other. The caption is again a bit of an essay with a method at the end that could be explained in the main text or in supplementary material.

We will provide a larger figure in the revised manuscript. Note that lines with the same colors are derived from the same experiment (warming level). Shading (and deviation from the x-axis) indicates which lines belong to the climatological mean and which to the 2018 ensemble.

Also here we give preference to an explanation below the figure in order for the reader to understand the figure without having to go through the text / supplement.

L395 Just to make sure, is the comparison with E-OBS done by the paper you are citing in the previous sentence? I was trying to find a graph that shows the comparison, but if it is in that paper, please clarify this in the sentence ("the authors found…" or something like that).

No, this concerns an evaluation of our simulations, which isn't included as figure, to limit the length of the manuscript. However, we can add a graph extending Fig. 1b to 2020 in the appendix.

L490-496 I do not think you need to state the obvious, plus you are not testing these impacts in this paper. You could save space by deleting this section.

These are the implications of our results, we prefer to leave this section in.

L510-L516 (optional) You could add spectrally nudged storylines to the list of options, it will allow for drought intensification studies, but not changes in dynamics (which you claim in L571-573 is plenty) with the benefit of a very small size ensemble (van Garderen & Mindlin 2022, https://doi.org/10.1002/wea.4185). However, since I am the author and it is for a region outside of Europe, feel free to ignore this comment.

Thanks for the reference, we'll add it to the introduction of the revised manuscript. In L510-516 we specifically discuss the 2018 event.

L521-525 Could it be that the discrepancy has anything to do with altering symptoms of climate change and not causes? See also my comment for L132-L134

This is unlikely, see our response to the comment on L132-134.

L543-545 In the absolute sense the referencing is correct, since the papers do mention analogues as well. However, storylines and analogues are not the same thing, and the emphasis of the paper cited is on storylines. Perhaps succinctly place the analogues in the context of storylines without doing another literature review (which you have already done).

In Sillmann et al. (2021) a storyline is described as: "*physically self-consistent unfoldings of past events, or of plausible future events, [which] have been proposed as a way of articulating the risk in such cases where we need to go beyond a purely probabilistic climate change perspective, with an emphasis on plausibility rather than probability*".

The future weather analogues of present-day events, as simulated in our study, fit this description. However, the reviewer is indeed right that we haven't explicitly stated this in our manuscript. We will place the future weather analogues in the context of storylines in the introduction of the revised manuscript, where we introduce our approach.

The comments I made are minor, and I am looking forward to seeing this paper published.

Best,

Linda van Garderen

---

## Author Response (AR1)

**Reply to Patrick Ludwig**

We thank Patrick Ludwig for his positive assessment of our manuscript and appreciate the suggested improvements. Please find our responses to the comments below in blue. Line numbers refer to the 'clean'/ 'tracked' revised manuscript.

Review of egusphere-2022-954 '*The 2018 west-central European drought projected in a warmer climate: how much drier can it get?*' by Aalbers et al.

**General comments:**

In their paper `*The 2018 west-central European drought projected in a warmer climate: how much drier can it get?´*, the authors investigate the impact of global warming on soil moisture drought severity for the year 2018 in west-central Europe. With this aim, pseudo-global warming (PGW) experiments with a regional model were performed, forced by three GCMs for three different global warming levels (+1.5K, +2K, +3K). Their results show, that under global warming the 2018 drought episode experiences strongly enhanced summer temperatures, but a fairly modest soil moisture drying response compared to the change in climatology as evaporation is already strongly moisture-constrained during present-day condition. In more general, the authors show that the drought risk in west-central Europe is strongly enhanced under global warming.

This work provides is a valuable contribution to our understanding of the consequences of global warming on extreme events (here drought) observed so far. The methods, techniques and the experimental setup applied in this study are sound and state of the art; the use of different GCMs (considering an ensemble mean for each GCM) to create the perturbations under global warming provides robust estimates of how global warming might modify the 2018 drought event. Beside some minor comments, this study is a very valuable contribution to NHESS.

**Minor Comments**

L62: What does the abbreviation SPEI stand for?

> SPEI is the Standardized Precipitation - Evaporation Index. We have removed SPEI in the revised manuscript, since *precipitation minus evaporation* in principle covers SPEI.

L135ff: Usually, global warming levels (GWL) are defined based on the pre-Industrial reference period (1850-1900). In the IPCC AR6 report, the global warming between 1850-1900 and 2011-2020 is estimate with 1.09K, thus meaning roughly GWL1. Based on the Paris agreement, the long-term temperature goal is to keep the rise in mean global temperature to well below 2K (GWL2) above pre-industrial levels, and preferably limit the increase to 1.5K (GWL1.5). To avoid any confusion, between the IPCC based and your GWLs (based on the 1991-2020 period), a short note might be helpful.

> We have added the following in the revised manuscript to emphasize this:

L. 163 ff / 187 ff

*Note that the global warming in the 1991-2020 period is 0.9°C with respect to pre-industrial period 1850-1900 (HadCRUT v5, Morice et al. 2021). We thus examine the impact of an additional 1.5°C, 2°C and 3°C global warming, which are projected to be reached within the 21st century under the RCP8.5 emission pathway in the GCM ensembles described below (see Tab. A2 for the specific time windows).*

L144ff: Can you comment briefly about the model quality of the 3 GCMs in comparison with observation? Is for example the temperature bias of the reference period you use (1991-2020) the same for all models? Or are the GCMs that already show enhanced global warming for this period in comparison with the observational record?

We have added in L.174 ff / 199 ff and adjusted the subsequent line:

*The simulated global mean temperature in the reference period in HAD and MPI are fairly close to the ERA5-reanalysis dataset (+/- 0.1°C, which is within one standard deviation of the global annual temperatures, see Tab. A1). EC-EARTH has a cold-bias in the reference period of 0.8°C compared to ERA5 for the global mean temperature. Since we use the GCM simulations only for the derivation of the perturbations (the present-day simulations are driven by ERA5), the impact of the biases in the GCMs is minimized. In the climate response there are large similarities between the three ensembles, all showing the north/south warming and drying gradient, but details like e.g. the response in the spatial pressure gradient and the shape of the vertical temperature response are different.*

We have added a table (Tab. A1) to Appendix A, which summarizes the characteristics of ERA5 and the three GCM-ensembles, including the global mean temperature and inter-annual variability (standard deviation) in the period 1991-2020.

L258: Should read '*over the British Isles*'

Thanks, we have changed this.

L393: The 2011 spring drought is not indicated in Figure 6b. Could you add this year to the graphs?

We have added the 2011-label to all graphs in Fig. 6b.

Figure1: $E_p$ is missing in the list of variables in (a); the orange lines for the observations (obs) for 2018-2019 are hard to see; consider to use a more striking color.

Thanks for noting, we have added $E_p$ in the caption, have adjusted the time series plots and changed the caption accordingly. For 2018 we have removed the shading that showed the inter-member spread of the simulations, and only show the individual ensemble members. The observations are now shown in red. Moreover, we have removed the shading that indicated the interannual variation in 1980-2017 in the observations so that the other information is more clearly visible. The comparison of observed and simulated inter-annual variability is now shown in the Appendix in a new plot Fig. C1a, similar to Fig. 1b, but for the climatology only.

Figure2: Consider moving the column with JFM as the first column; would better reflect the course of the year.

We chose to start with April as the start of a hydrological year, but we don't have a strong preference. We have adjusted Fig. 2 and now show JFM in the first column.

**Reply to Linda van Garderen**

*We thank Linda van Garderen for her thorough reading, constructive comments and the overall positive assessment of our manuscript. In the following, we address the comments point-by-point. Line numbers refer to the 'clean'/ 'tracked' revised manuscript.*

General remarks

The paper is a comprehensive study of the 2018 west-central European drought focussing on soil wetness deprivation, and what that drought would have been like in a world with different levels of climate change using the pseudo global warming method and analogues. The study is technically rich, as the analysis is done with different global climate models for input to the regional model providing a cross-check for the results, as well as content-wise rich with comparison to other drought events and a climatological test of droughts in this region and elaborate placement of their findings in existing literature.

I find this research very relevant as it produces new insights in the attribution of European droughts. The analysis of the drought is well executed and technically sound, including temperature, precipitation and evaporative demand. Limitations to the analysis are also reflected on accurately. The paper is lengthy, though, and could benefit from a reduction in size (I have tried to provide some small suggestions for this in the minor comments below). The English language is of high quality and the paper reads well.

Detailed remarks

You have used "(not shown)" six times in this paper (L221, L235, L276, L279, L329, L340). This is too often and becomes a hindrance since it implies a demand of trust. Either show what you claim, rephrase, reference to a paper that proofs a comparable outcome to the one you found, or delete the claim. Please try to reduce the "(not shown)" to one or two times. Since you have a large appendix already, you might consider placing some of these aspects in the supplementary material to prevent the paper from becoming larger.

*Thank you for pointing this out. We have reduced the number of not showns in the revised manuscript by adding a figure in the SI for some of them. Below each of the original six not showns is commented:*

*L221: As a consequence, the sensible heat flux strongly increases (not shown), ..*

*This concerns the increase in sensible heat flux during the summer of 2018, which logically follows from the increase in evaporative demand (shown), and decrease in evaporation (shown) and soil moisture (shown). In our opinion, here no additional figure is needed.*

*L235: For deeper soil layers the winter precipitation is insufficient to fully replenish the soils to normal levels, and the anomalously dry conditions persist throughout 2019 (not shown).*

*Since the focus of this manuscript is on the top 1 m of the soil, we decided to limit the figures to variables that are directly relevant for the top 1 m of the soil. However, readers may remember that groundwater levels were lower than normal throughout winter and spring 2019. To show that this is the case in our simulations as well we have added Fig. C1b in a new Appendix C Present-day simulations.*

Fig. C1b is similar to Fig. 1b, but shows the time series from March 1$^{st}$ 2018 – December 31$^{st}$ 2020, and includes the evolution of runoff and the soil wetness index in the total soil layer. *not shown* in former L235, now L264/L290 is replaced by *see Fig. C1b*.

L276: This feature of the response is amplified by reduced snowmelt in spring and a larger fraction of precipitation falling as rain in autumn (not shown).

We have adjusted this sentence in the revised manuscript as follows (L314/L345):

*This feature of the response is amplified by a larger fraction of precipitation falling as rain in autumn (due to higher temperatures) and reduced snowmelt in spring (less snow to melt).*

L279: In summer and autumn, the soil moisture availability in deeper layers and runoff decrease as well (not shown).

We have added a new Fig. D3 in Appendix D (former Appendix C), similar to Fig. 3 but for two additional variables: we show the soil wetness index in the total soil layer ($SWI_{0-2.9m}$) and runoff (Q). We have extended former Fig. C4 and C6 (now Fig. D4 and Fig. D5), showing the response for MPI and HAD, with these variables as well. *Not shown* has been replaced by *see Fig. D3*.

L340: Percolation to deeper soil layers and runoff decrease in this period as well (not shown).

Actually this is shown later in the manuscript, in Fig. 5b, and in the new Fig. D3, see our previous response. *not shown* is replaced by *see Fig. D3 and Fig. 5b, discussed below*.

Not all graphs are easily readable. I reviewed using a print-out and could not read Figure 1 and Figure 5a, and to some extent Figure 3. I have put a more detailed explanation below, but would encourage giving figures a check-over for readability.

We're sorry that you couldn't easily read some of the figures in the printed version of the manuscript. Apart from the color in figure 1b the other reviewers did not comment on the figures, so we trust that larger (online) versions of the figures are clear. We provide full-width versions of the multi-panel plots in the revised manuscript. Moreover, we have adjusted Fig. 1b, Fig. 3 and Fig. 5a. See our response to the detailed comments on these figures below.

Minor comments

L52-L56 This is 1 very long sentence. Please shorten for ease of understanding.

L52ff/L52ff now reads:

*Although the probability of heat waves in this region is demonstrated to have increased in response to anthropogenic climate change (Stott et al., 2004, Vogel et al., 2019, Vautard et al., 2020), the attribution of extreme drought events is more complex (Trenberth et al., 2014). Independent drought events are scarce, owing to their long timescale and large spatial scale, which hampers the derivation of robust statistics. Moreover, the processes contributing to wide-spread drought conditions are not easily disentangled.*

L67-L69 This sentence could use a reference to back up the claims. For the Mediterranean precipitation research, you could cite Zappa et al., 2017 (http://dx.doi.org/10.1175/JCLI-D-16-0807.s1).

> The response pattern over Europe described in L67-69, including the response over west-central Europe (L70-71) is shown in the references in line L71. We have moved these references one line up in the revised manuscript to make this more clear.

L99-L100 Is your study trying to provide something like the 'common framework' published by Shepherd et al., 2016? (http://dx.doi.org/10.1007/s40641-016-0033-y)

> *L99-100: Additionally we evaluate the position of this 2018 event in the 1980-2020 period, both for present-day and for future conditions under a single warming level.*

> Shepherd (2016) discusses extreme event *attribution* to climate change, and we wouldn't suggest that we do anything like that (a 41-year simulation period is way too short for attribution). However, simulating a future weather analogue not only of the extreme 2018 event, but of all summers in the 1980-2020 period, provides context and a better understanding of the changes we find for the extreme summer of 2018.

> No revisions were made based on this question.

L132-134 You are altering a large number of variables to create the counterfactual worlds. Could you argue why it is acceptable to meddle with the model to such a large extent without loosing physical self-consistency? In essence, you are altering the consequences of climate change (T, P, SH, etc.) instead of (only) the causes of climate change (GHG, SST, etc.). I am aware that the method requires this to create a counterfactual world, but I would like to hear the validation in 1 or 2 short sentences (could be in discussion you find that a better fit).

> The strength of the application of a regional climate model for this purpose is that the simulations are physically consistent in the interior of the model domain by design. The perturbations itself are derived from GCM simulations and are added to *all* state variables, again to ensure physical consistency. Moreover, the perturbations are fairly small (apart from temperature) and are smoothly varying in space and time. See also e.g. Prein et al. 2016 and Shepherd et al. 2018.

> Of course, we built our PGW simulations on the large-scale circulation in the present-day period, and it may be that the frequency of e.g. blocking conditions changes in a globally warmer world. However, dynamical changes are highly uncertain, and there is no physical argument against the plausibility of the present-day circulation types under globally warmer conditions.

> We have added to the methods section:

> L164 ff / L189 ff

> *The physical consistency of the PGW simulations is ensured by perturbing all state variables with a consistent set of perturbations derived from GCM projections. Moreover, the perturbations are (apart from temperature) fairly small and smoothly varying in space and time. In the interior of the RCM domain, simulations are physically consistent by design.*

L145-146 Please add the resolutions of the individual EC-EARTH v2.3, HadGEM2-ES and MPI-ESM-LR GCM's.

> We have added a new table (Tab. A1) to Appendix A, which summarizes the characteristics of ERA5 and the three GCM-ensembles, including the resolutions. We have adjusted the format of former Tab. A1 (now Tab. A2) to match the new Tab. A1.

L169-173 The reader might benefit from adding an equation to show this second step as well.

> We have added equations 2 and 3, and have adjusted the text accordingly (adjustments in red):
>
> L197 ff / L223 ff
>
> *We express the drought severity in terms of the drought deficit volume ($D_S$, unit mm d), which integrates drought duration ($\tau$, d) and drought intensity ($D_I$, mm), comparable to e.g. Yevjevich (1967) and Brunner et al. (2019). It is calculated as the accumulated difference between $\theta_{5th}$ and $\theta$ over the drought episode (Eq. 2). The analysis is based on a time series with daily values, so that the time step $\Delta t$ in Eq. 2 is 1 day, and the index i varies from the first day (i = 1) to the last day (i = n) of the drought episode, with $\tau = n$ days. The drought intensity is defined as the drought deficit volume divided by the drought duration (Eq. 3).*
>
> $$D_S = \sum_{i=1}^{n}(\theta_i - \theta_{5th,i})\Delta t \qquad (2)$$
>
> $$D_I = \frac{D_S}{\tau} \qquad (3)$$

Figure 1 The b panels are not readable at all, please change colours and either make the graph bigger or lines a little thinner. Ep is not mentioned in the caption. The legend in the b graphs is incomplete (shading is not mentioned, for instance).

> We have adjusted the figure as follows: the legend of the maps in Fig. 1a is now positioned below the subplots to create more space for the time series plots. The legend of the time series plot is now positioned above the first subplot and has been extended. We have replaced the 95th/5th percentile dashed line with shading, to reduce the number of different line styles. For the 2018 ensemble we removed the shading and only show the individual ensemble members. For the observations in the time series plots, we have changed the color to red, and we have removed the 1980-2017 inter-annual variability. The comparison of observed and simulated inter-annual variability is shown in a new plot C1b in the Appendix, similar to Fig. 1b, but for the climatology only.
>
> The caption now reads (changes in red):
>
> *Figure 1: The 2018 drought episode. a) Maps of the simulated seasonal mean anomaly with respect to 1980-2017 in 500 hPa geopotential height (contours, in m) and (top to bottom) near surface-temperature (T), precipitation (P), potential evaporation ($E_p$), evaporation (E) and top 1 m soil wetness index ($SWI_{0-1m}$), for April – June (AMJ), July – September (JAS) and October – December (OND). Data are masked over sea for visibility. The west-central European river basins are marked in black. b) Time series of the simulated basin-mean (top to bottom) temperature, precipitation, (potential) evaporation*

*and top 1 m SWI for January 2018 – March 2019 and the 1980-2017 climatology. Thin black lines show the 2018REF ensemble members. The thick grey line and dark and light grey shading depict respectively the 1980-2017 mean, 25th-75th and 5th-95th percentiles. Observed (E-OBS v20.0) temperature and precipitation are shown along for 2018 (red line) and the 1980-2017 mean (pale red line). Time series are smoothed with a 14-day running mean.*

In the text we added in L229/L255:

*The climatological mean, 5th and 95th percentile thresholds in observed and simulated temperature and precipitation are shown in Fig. C1a.*

L203 I do not believe the pressure anomalies are shown? Or are they the once in the supplementary material? Please refer if that's the case or show what you are claiming or reference to a paper that shows it.

We show the 500 hPa geopotential height anomalies (contours in Fig. 1). We have replaced 'pressure anomalies' with 'the 500 hPa geopotential height anomalies' and refer to Fig. 1 in the revised manuscript:

*The 500 hPa geopotential height anomalies in (late) spring (AMJ) and summer (JAS) clearly co-occur with the large positive temperature anomalies and high precipitation deficits (contours and shading in Fig. 1a, top two rows).*

L261 The reader could benefit from adding which colour line to look at when referring to Figure 3a. Also, to add which colour to look at with "mean response is rather large…"

We have added this information as follows:

*Averaged over the river basins the near-surface temperature response varies between +1.4°C in May and +2.6°C in August (Fig. 3a, black line). Note that the inter-annual spread around the 1980-2017 mean response (grey box and whiskers) is rather large.*

L266 "increasing cloud cover…" is not shown anywhere, or do you mean an interpretation of solar radiation in Figure 3b? Please clarify.

The increase in solar radiation in JJASO indeed results from decreases in cloud cover (and humidity). The response in cloud cover is not shown, but is not critical for the results. L305 / L335 now reads:

*This co-occurs with nearly constant relative humidity in late autumn, winter and early spring, and decreases in relative humidity in JJASO (Fig. 3d). Net surface solar radiation increases in JJASO (Fig. 3b), when cloud cover and relative humidity decrease.*

L282-L283 Figure C1 and C2 show vastly different GPH z500 patterns. Should this be mentioned or explained? Is this of significance for the analysis?

We describe this briefly in the methods section (2.2). The GCMs indeed differ in their regional response pattern. That is the reason for using perturbations derived from three different GCMs. No adjustments were made in response to this question.

Figure 3 The legend is incomplete, please also mention what the shading etc. stands for. The caption is a bit of an essay. I would suggest placing the method part that right now is at the end of the caption, including the equation, to the main text or supplementary material. From a print the graph is difficult to read, which is a pity since it shows some essential and interesting things. You could either make it bigger or reduce the thickness of the median (black or red).

> We have adapted Figure 3 as follows: we have increased the height of the subplots; the lines of the 2018 mean and 1980-2017 median are thinner; the 2018 box is more narrow; 2019 and 2020 have an unique color; we have replaced the legend by a more elaborate legend above the graphs.

> Moreover, to shorten the caption, we have removed the last line and now describe the calculation of the inter-member variability in the methods section:

> L212 ff / 237 ff:

> *2.5 Inter-member variability*

> *The inter-member variability (IMV) of the 2018 ensemble simulations is measured by the standard deviation (σ). The ensemble members of the present-day (2018REF) and PGW (2018+nK) simulations are independent, since the simulations are performed for two separate time slices (opposed to continuous simulations). Therefore, the inter-member variability of the difference between PGW and REF ($\sigma_\Delta$) is calculated from the standard deviation of the present-day ($\sigma_{REF}$) and PGW ($\sigma_{PGW}$) simulations:*

$$\sigma_\Delta = \sqrt{\frac{\sigma_{REF}^2 + \sigma_{PGW}^2}{2}} \qquad\qquad (4)$$

Figure 5a The lines are almost all the same colour, and to thick to interpret. There are even two lines that are both red. Please update the graph, make the lines thinner or the graph bigger, and choose colours that are further apart from each other. The caption is again a bit of an essay with a method at the end that could be explained in the main text or in supplementary material.

> As for Fig. 1b, we have replaced the lines indicating the mean climate and 95[th] or 5[th] percentiles with a shaded envelope, to reduce the number of line(style)s. Furthermore we have enlarged the figure for better visibility. A slightly brighter color is selected to depict the 2018+1.5K ensemble. The colors for 2018+2K and 2018+3K are kept as is, since this color range allows for an intuitive interpretation (stronger global warming when moving from yellow to bright red to dark red). Note that the inter-member variability is relatively large in some periods, while differences between the ensemble means are small. A different color set wouldn't improve the visibility of the individual ensembles. Concerning the caption: here we give preference to an explanation of the method below the figure in order for the reader to understand the figure without having to go through the text / supplement.

L395 Just to make sure, is the comparison with E-OBS done by the paper you are citing in the previous sentence? I was trying to find a graph that shows the comparison, but if it is in that paper, please clarify this in the sentence ("the authors found…" or something like that).

No, this concerns an evaluation of our simulations, which isn't included as figure since 2019/2020 wasn't the main focus and to limit the length of the manuscript. However, to prevent a new '*not shown*' we have added an extended version of Fig. 1b to appendix C (Fig. C1b), showing the period March 2018 – December 2020. We refer to this figure in L434/470 (former L395).

L490-496 I do not think you need to state the obvious, plus you are not testing these impacts in this paper. You could save space by deleting this section.

These are the implications of our results, we prefer to leave this section in.

L510-L516 (optional) You could add spectrally nudged storylines to the list of options, it will allow for drought intensification studies, but not changes in dynamics (which you claim in L571-573 is plenty) with the benefit of a very small size ensemble (van Garderen & Mindlin 2022, https://doi.org/10.1002/wea.4185). However, since I am the author and it is for a region outside of Europe, feel free to ignore this comment.

Thanks for the reference, we have added it to the introduction of the revised manuscript. (In L510-516 we specifically discuss the 2018 event.)

L88 ff / L96 ff:

*Storylines in the form of future analogues of heatwaves and droughts have previously been constructed with e.g. spectral nudging of GCMs (Rasmijn et al., 2018, Wehrli et al., 2020, Van Garderen and Mindlin, 2022), and by selecting events from different warming periods in a very large GCM ensemble (Van der Wiel et al., 2021).*

L521-525 Could it be that the discrepancy has anything to do with altering symptoms of climate change and not causes? See also my comment for L132-L134

This is unlikely, see our revisions in response to the comment on L132-134.

L543-545 In the absolute sense the referencing is correct, since the papers do mention analogues as well. However, storylines and analogues are not the same thing, and the emphasis of the paper cited is on storylines. Perhaps succinctly place the analogues in the context of storylines without doing another literature review (which you have already done).

The future weather analogues of present-day events, as simulated in our study, fit the description of an (event-based) storyline in Shepherd et al. (2019) / Sillmann et al. (2021). However, the reviewer is indeed right that we hadn't explicitly stated this in our manuscript. In the revised manuscript we have adjusted paragraphs 5 and 6 of the introduction. We first introduce the storyline approach and future weather analogues, subsequently refer to different methods to create storylines (see previous comment), and then elaborate on the PGW approach employed in our study.

L 79 – 87 / L 81 – 95 now reads:

*In this study the contribution of global warming to the increase in drought severity and frequency is being addressed by projecting the 2018 drought, as well as the entire 1980-2020 historical period, in a globally warmer world. This is an implementation of the so-called storyline approach (Hazeleger et al.,*

*2015, Shepherd, 2018, Shepherd et al., 2019, Sillmann et al., 2021), a storyline being defined as a 'physically self-consistent unfolding of a past event, or of a plausible future event or pathway' (Shepherd et al., 2019). Instead of analyzing large ensemble simulations to derive changes in the probability of extreme drought events, we construct plausible future drought events as analogues of extreme climate events that have actually occurred in the current climate (Hazeleger et al., 2015). As such, changes in droughts in response to global warming can be directly related to the real world events and their societal impact, which make the results very tangible and therewith useful for climate change communication.*

Note that L 88 – 106 / L 96 – 114 have significantly been adjusted as well.

The comments I made are minor, and I am looking forward to seeing this paper published.

Best,

Linda van Garderen

**Reply to anonymous Referee #3**

We thank the reviewer for the positive assessment of our manuscript. Please find our reply to the minor comments below in blue. Line numbers refer to the 'clean'/ 'tracked' revised manuscript.

General Overview:

The authors investigate the impact of global warming on soil moisture drought severity in west-central Europe by employing pseudoglobal warming (PGW) experiments.

The manuscript is well written and the methodology is sound and the results are novel. In my opinion, the manuscript can be accepted after minor revision mentioned below.

The authors mention "future weather analogues" in the manuscript but actuallaly no analogue metdology is applied (e.g. Sánchez-Benítez et al., https://doi.org/10.1016/j.wace.2019.100238 ). I would suggest the authors to change the wording on "weather analogues" throughout the manuscript.

> We agree that we apply a different method than Sánchez-Benítez et al. However, the future weather analogues as created by the PGW simulations fit the description of future weather analogues introduced by Hazeleger et al. 2015. We have adjusted the section where we introduce our approach (this includes adjustments in response to the comment of reviewer Linda van Garderen on L.543-545).
>
> L.79 ff / L.81 ff
>
> *In this study the contribution of global warming to the increase in drought severity and frequency is being addressed by projecting the 2018 drought, as well as the entire 1980-2020 historical period, in a globally warmer world. This is an implementation of the so-called storyline approach ( Hazeleger et al., 2015, Shepherd, 2018, Shepherd et al. 2019, Sillmann et al., 2021), a storyline being defined as a 'physically self-consistent unfolding of a past event, or of a plausible future event or pathway' (Shepherd et al., 2019). Instead of analyzing large ensemble simulations to derive changes in the probability of extreme drought events, we construct plausible future drought events as analogues of extreme climate events that have actually occurred in the current climate (Hazeleger et al., 2015). As such, changes in droughts in response to global warming can be directly related to the real world events and their societal impact, which make the results very tangible and therewith useful for climate change communication.*

Despite the methodology is sound, I had difficulty to follow section 2.2. Can the authors re-write section 2.2.

> We went through the methods section and have restructured and reformulated parts of the text, especially in the subsection on the PGW simulations, see L. 146 – 166 / L. 161 - 191. Moreover, we elaborate a bit more on the PGW approach and interpretation of the simulations in the introduction, so that the approach is clear right from the beginning. For that reason we moved the part on the interpretation of the simulations that

was previously in the methods section (L180-183 in the revised manuscript with tracked changes) as well as the part where we describe other applications of the PGW approach in the discussion (L541-545) to the introduction (see L90-106 / L98-114). Hopefully this makes it easier to follow section 2.2.

Why the authors exclude 2018-2020 from the climatology?

When analyzing the exceptionality of a period or event, it is not uncommon to exclude that period from the climatology to avoid selection bias. However, in our application, the effect of including 2018-2020 in the climatology is fairly small, and is very unlikely to affect the conclusions. Since it is fairly common, we don't think an explanation is required in the manuscript.